# Combined mutations of *hoxa13a, hoxa13b*, and *hoxd13a* lead to structural shifts in zebrafish soft fin rays providing insight into spiny ray evolution

Jordan Corcoran [1], Hailey Quigley[1], Qingming Qu[1,2], Marie-Andrée Akimenko [1]*

1 Department of Biology, University of Ottawa, Ottawa, Ontario, Canada, 2 State Key Laboratory of Cellular Stress Biology, School of Life Sciences, Xiamen University, Xiamen, Fujian, China

* marie-andree.akimenko@uottawa.ca

## Abstract

The zebrafish *hoxa13a, hoxa13b*, and *hoxd13a* genes have been shown to be essential for proper pectoral fin patterning. Combined mutations in these genes cause an elaboration of the pectoral fin endoskeleton, and a size reduction of the rays. In this study, we further examine the impact of partial deletions in these genes on the fin exoskeleton. Using morphological and micro-CT scan analyses, we found that rays of all fins of triple *hox13* mutants are strongly affected, except for the caudal fin that is much less perturbed. Rays are shorter and thicker than wildtype rays, and present a loss of joints, bifurcations, and actinotrichia. Altogether, they lose many of the typical soft ray features and resemble more to the spiny rays of acanthomorphs. In these species, actinotrichia are present in spiny rays but organized differently than in soft rays, and spiny rays develop in the absence of *hoxa13a/b* expression. Gene expression analysis of triple *hox13* mutant zebrafish larvae shows an expansion of the *alx4a* expression domain relative to the size of the dorsal and anal fin primordia and a reduction in *grem1b* expression that are reminiscent of the differences observed in acanthomorph spiny versus soft rays. Using various combinations of genotypes, *hoxa13a* and *hoxa13b* were found to be more important for normal soft ray formation than *hoxd13a.* In conclusion, our results demonstrate the importance of the *hox13* paralogs for proper soft ray development and suggest a lack of *hox13* expression could be important for their transformation into spiny rays.

## Author summary

Fins of bony fish present incredible diversity in the shape, structure, and number of the fin rays. Among teleosts, acanthomorph fish have sharp spiny rays in their fins that are thought to serve as a defense mechanism from predators. These spines evolved in teleost fish around 133–150 million years ago, but how exactly they evolved, and the genetic mechanisms involved in the appearance of

**Data availability statement:** All relevant data are within the manuscript and its Supporting information files.

**Funding:** This work was supported by grants from the Canadian Institute of Health Research Council (PJT 166139) and the Natural Sciences and Engineering Research Council of Canada (RGPIN-2024-06801) to M.A.A. and the Natural Science Foundation of Xiamen, China (3502Z202473009) to Q.Q. The funders had no role in the study design, data collection and analysis, decision to publish, or preparation of the manuscript.

**Competing interests:** The authors have declared that no competing interests exist.

this structure are not totally clear. Zebrafish (*Danio rerio*), a common fish model amenable for genetic studies, do not normally contain spines in their fins and only have soft rays formed by the stacking of short bone segments. In our study, we show that genetically modified zebrafish develop fin rays that show close similarities to spines, including shorter length, lack of segments and bifurcations, and thicker bone that are all seen in spines but not in soft rays. Our mutant zebrafish have deletions in the *hoxa13a, hoxa13b,* and *hoxd13a* paralogous genes, that code for transcription factors with the ability to control the expression of downstream target genes. Our findings show that modifications to the expression of these *hox* genes could have influenced the evolutionary appearance of spines in fish fins.

## Introduction

The zebrafish fin rays are composed of two biconcave hemirays made of bone forming by intramembranous ossification that encase the mesenchymal tissue, connective tissue, blood vessels and nerves of the ray [1–3]. These hemirays are made of bone segments that are separated by fibrous joints and are held together by ligaments [3]. Each ray, except for the lateral-most rays of the caudal fin, and anterior-most rays in the other fins, bifurcate into two sister rays, sometimes multiple times along the proximal-distal axis [3]. At the distal tips, each ray contains bundles of rigid fibres known as actinotrichia [1,3]. These characteristics define the soft rays. With teleosts being the largest and most diverse class of vertebrates [4], this high diversity leads to many morphological differences in fin structure such as the overall shape and lobes of the fins, and the number and length of rays in each fin [5,6]. A specific example is shown between two groups of the teleost class, Cypriniformes and Acanthomorphs [6]. Cypriniformes, with a few exceptions, have only soft rays [6]. In contrast, the dorsal and anal fins of most acanthomorphs also have spiny rays in the anterior domain of their fins [7,8]. Unlike soft rays, the spiny rays do not contain fibrous joints or bifurcations and are generally shorter in length [8,9]. In addition, the hemirays of the spines are fused at the anterior end instead of being separated and are more heavily ossified than the soft rays [10]. The distal tip of spiny rays also ends in a sharp point, which is suggested to serve as a defense mechanism against gape-limited predators [10,11]. While spiny rays are a key defining feature of the acanthomorphs, they have also evolved convergently in several other species such as carps, sturgeons, and catfish [11,12]. Acanthomorphs evolved more recently, around 133–150 MYA [13], compared to the Cypriniformes which evolved around 194 MYA [9], however the mechanism of how these spines arose, and why we see these differences in fin ray structure in certain species remains to be fully understood. During early development of the anal and dorsal fins of *Astatotilapia burtoni*, an acanthomorph fish, anterior spiny and posterior soft rays appear similar but become distinguishable as the unique characteristics of each ray type begin to develop [9]. Phenotypic differences between ray types are accompanied by distinct transcriptional signatures in their respective

domains during dorsal and anal fin development [9]. The posterior soft ray domain side shows exclusive expression of *hoxa13* genes whereas the spiny ray domain contrastingly lacks *hoxa13* expression and shows expression of *alx4a* instead [9]. In adult zebrafish, *alx4a* has been shown to be expressed in the osteoblasts and fibroblasts of anterior rays of the adult fins [14], but the expression of this gene in the early primordia of the dorsal and anal fins for zebrafish has not yet been examined.

Although it is not known how exactly spiny rays emerged during evolution, shifts in *hox* expression could potentially play a role in this structural change. The *Hox* family of genes are organized into clusters in the vertebrate genome [15] and code for homeodomain transcription factors that are highly conserved throughout all vertebrates [16]. Mammalian species have four *hox* clusters (*HoxA, HoxB, HoxC,* and *HoxD*). However, zebrafish have seven clusters (*hoxaa, hoxab, hoxba, hoxbb, hoxca, hoxcb,* and *hoxda*) due to a genome duplication event that occurred in ancient teleost fishes [17] around the early-middle Permian era [18] followed by the subsequent loss of the *hoxdb* cluster. As originally described in the *Drosophila* cluster [15] and later in other species [19], *hox* gene organization in the genome is colinear to their respective domains of action during embryonic development. HOX transcription factors specify morphogenetic fates in cells in each position that they are expressed in [20]. Loss of function mutations in *Hox* genes have been shown to cause homeotic transformations of body structures in many organisms including *Drosophila* [21,22], chickens [23,24], and mice [25–28]. The *HoxA/D13* genes specifically have been shown to be essential for patterning the distal part of the limb in many tetrapod species, including in mice [29] and humans [30]. Less is known about the role of these genes on the zebrafish fins, although *hoxa13a, hoxa13b,* and *hoxd13a* paralogs have been shown to be essential for the proper development of the pectoral fin endoskeleton through the observation of triple homozygous mosaic mutants [31]. During zebrafish caudal fin regeneration, *hoxa13a, hoxa13b*, and *hoxd13a* are expressed at the tips of the regenerating rays, highlighting the probable importance of these genes for fin ray growth [32]. In addition to the blastema, *hoxa13a* has been shown to be expressed in preosteoblasts and committed osteoblasts as well as in all joints of the regenerating caudal fin rays. An upregulation of *hoxa13a* expression was observed in presumptive joint cells suggesting its potential role in the initiation of joint formation [32].

In this study, we generated a stable line of triple homozygous *hoxa13a* [-/-], *hoxa13b* [-/-], *hoxd13a* [-/-] mutant fish (referred to as triple *hox13* mutants) and analyzed the ray structure of each fin. These combined mutations cause severe defects including a loss of joints, bifurcation, and actinotrichia in rays of all fins except for the caudal fin. These defects in combination result in affected rays that exhibit characteristics similar to spiny rays instead of soft rays. We also characterized double homozygous, single heterozygous compound mutants for these genes to evaluate the individual contributions of each of these *hox13* genes to the ray phenotype. Overall, *hoxa13a* and *hoxa13b* demonstrate greater importance to ray joint, bifurcation, and actinotrichia formation compared to *hoxd13a* and also allow for increased length of the rays. In addition, we demonstrate that triple *hox13* mutants have smaller dorsal and anal fin primordia during larval development. In summary, these results demonstrate that combined mutations in these genes may contribute to a structural transformation of the soft rays that makes them appear more similar to spiny rays revealing the functional involvement of *hoxa13a, hoxa13b*, and *hoxd13a* in the exoskeleton patterning of all fins except the caudal fin.

## Results

### The combinatorial contribution of *hoxa13a, hoxa13b,* and *hoxd13a* to fin ray phenotype

Using CRISPR-Cas9, we induced deletion mutations in the zebrafish genome within the *hoxa13a, hoxa13b,* and *hoxd13a* genes (Figs 1A and S1B). Two gRNAs were synthesized and injected for each gene to delete portions of the coding sequence, including the homeobox for *hoxa13a* and *hoxd13a* (S1 Table). Single homozygous deletions in these genes do not cause any severe morphological defects aside from small reductions in the length of the fin rays (similar to published results in Nakamura et al. [31]). The viable single mutants were then crossed with one another to obtain lines of double and triple mutants for the *hox13* genes. All lines are viable, but present with variable phenotypic defects of their fins as

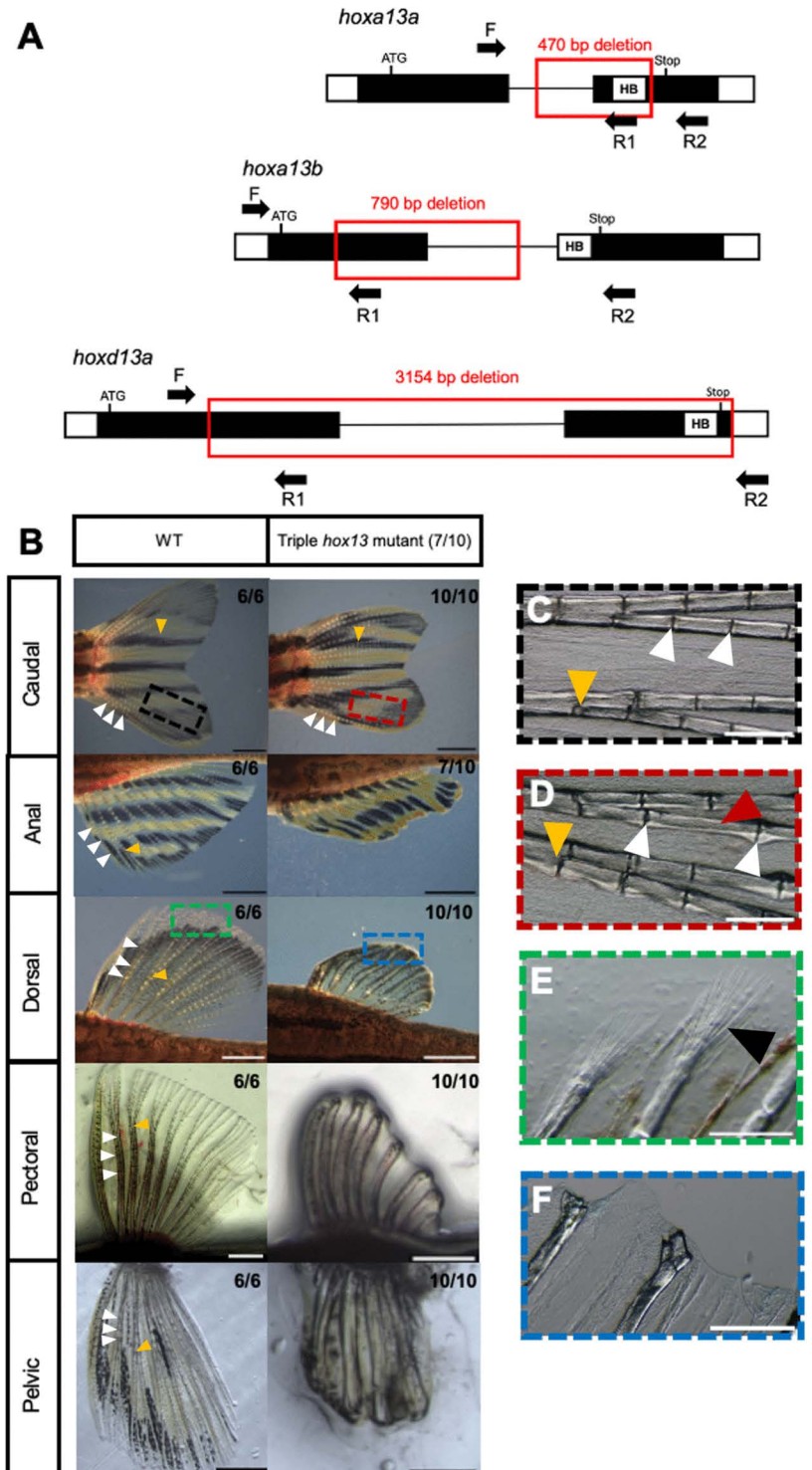

**Fig 1. Triple homozygous *hox13* mutants have severe defects in ray patterning in all fins except the caudal fin. A)** Schematic representation of CRISPR-Cas9 induced deletions in *hoxa13a, hoxa13b,* and *hoxd13a*. Portions of the genes deleted in the mutants are shown within the red box. Translation start and stop sites are represented as ATG and Stop respectively. Locations of genotyping primers are indicated as black arrows as either F (forward) or R (reverse). Homeobox sequence locations are represented as HB. The black boxes represent exons, and the white boxes represent untranslated region. **B)** Caudal, dorsal, anal, and paired fins of wildtype siblings (n = 6) and triple homozygous *hox13* mutant fish (n = 10). Phenotypes

were consistent with the exception of n = 3/10 anal fins in the triple *hox13* mutant, which show irregular joint spacing in the anal fin (S1C Fig). Yellow arrowheads show examples of bifurcation, and white arrowheads show examples of joints present in all fins of wildtype fish as well as in the caudal fin of triple *hox13* mutants but absent in the pectoral, pelvic, dorsal and most of the anal fins of the triple *hox13* mutants. Scale bars, 2mm for caudal and anal fins, 1.5mm for dorsal fins, 1mm for pectoral fins, 0.5mm for pelvic fins. **(C-D)** Enlarged images of triple *hox13* mutant and wildtype sibling caudal fin rays corresponding to the boxed area shown in **B**. Joints are regularly spaced in rays of wildtype fish **(C)**. White arrowheads indicate joints and yellow arrowheads indicate bifurcations. The red arrowhead in D indicates a long bone segment with abnormal joint spacing in the mutant. Scale bars, 100 µm. **(E-F)** Enlarged images of the triple *hox13* mutant and wildtype sibling fin ray tips are presented to demonstrate the lack of actinotrichia fibers at the tips of the mutant rays. Scale bars, 100 µm.

described below. Only the triple homozygous *hox13* mutants are sterile and must be obtained by heterozygous crosses as a result. *In vitro* fertilization was attempted with triple *hox13* mutant males to test for fertility, but the sperm collected was unable to fertilize wildtype eggs. The sperm itself was not analyzed further. In the case of the females, they were unable to release eggs when paired with wildtype males. When *in vitro* fertilization was attempted, eggs were unable to be released from any of the triple *hox13* mutant females. Upon further analysis of the females, the urogenital pore of the triple mutant appears malformed and shortened which may contribute to the female inability to release eggs (S1F and S1G Fig). The urogenital pore structure of these triple mutants was not further analyzed, but it has recently been shown that a fused hindgut and pronephric duct is already observed at 6dpf in other *hoxa13a* $^{-/-}$, *hoxa13b* $^{-/-}$, *hoxd13a* $^{-/-}$ mutants [33].

Analysis of the pectoral fin endoskeleton in triple homozygous mutants of *hoxa13a*, *hoxa13b*, and *hoxd13a* reveals an increased number of distal radials, a phenotype consistent with prior observations of mosaic *hox13* mutants and homozygous mutants of *hoxa13a* and *hoxa13b* reported by Nakamura et al. [31] 4/4 observed triple homozygous *hox13* mutant have 10 distal radials in the pectoral endoskeleton compared to the 7 distal radials of the wildtype siblings (S2A-B and S3A Figs). There can at times also be a reduction in the number of fin rays in this fin in the triple *hox13* mutant, where 9 rays can be observed instead of 10 (S2E and S3B Figs). In the pelvic fin of this mutant, there are no changes in the number of distal radials (S2C, S2D, and S3A Figs), however there is a decrease in the number of rays, where 5 rays are seen as opposed to 7 in the wildtype (S2F and S3B Figs). In the triple *hox13* mutant dorsal and anal fins, the number of rays and radials can be reduced by 1 compared to wildtype, which happens with approximately 50% frequency (S2G-J, S3A, and S3B Figs). However, there is no discrepancy between the number of radials and number of rays in a given dorsal or anal fin. Furthermore, the reduced number of fin rays and radials does not always happen in the same individual for both fins. Of the 12 mutants observed, 4 out of the 12 fish have a reduced number in both the dorsal and anal fins, 3 fish have exclusively a reduction in the dorsal fin, and 2 fish have exclusively a reduction in the anal fin. This indicates that the ray and endoskeletal reduction is not 100% penetrant. Finally, no changes in the endoskeleton or the number of rays were observed in the caudal fin of the triple mutants (S2K and S2L Fig). Exact fin ray and distal radial counts are shown in S3 Fig for wildtype and triple *hox13* mutants.

Besides the number of rays differing in the triple mutant fins, the structure of the rays is affected as well. All fins of the triple homozygous *hox13* mutants are affected to varying degrees when compared to wildtype siblings, but all demonstrate a reduction in length compared to their wildtype siblings. The most severe reductions in length are observed in the paired fins, where pectoral and pelvic fin rays are only approximately 32% and 22% as long as their wildtype counterparts, respectively (Fig 2). Interestingly, these rays lack joints and bifurcations, two characteristics of the soft rays of zebrafish. Furthermore, there is a lack of visible actinotrichia fibers protruding from the tips of the rays (Figs 1F and S1D). Consequently, instead of tapering at their distal tips, the rays of the triple mutants end abruptly and are often curled at the tips (Figs 1F and S1D). In contrast to the other fins, the combined mutations in the *hox13* genes have less of an effect on the caudal fin. A significant length reduction of about 25% is seen in the rays of each lobe, but joints, bifurcations, and actinotrichia (except for the lateral-most rays) remain present in this fin (Fig 2H-J). The only defect observed in the joints of these fins was that in the triple *hox13* mutant caudal fin rarely have longer bone segments which are approximately twice as long as the other neighboring bone segments within the same ray (Fig 1D). This however is relatively rare as an average of 3 longer segments per fin (stdev = 1.56) were observed in the 10 triple *hox13* mutants analyzed.

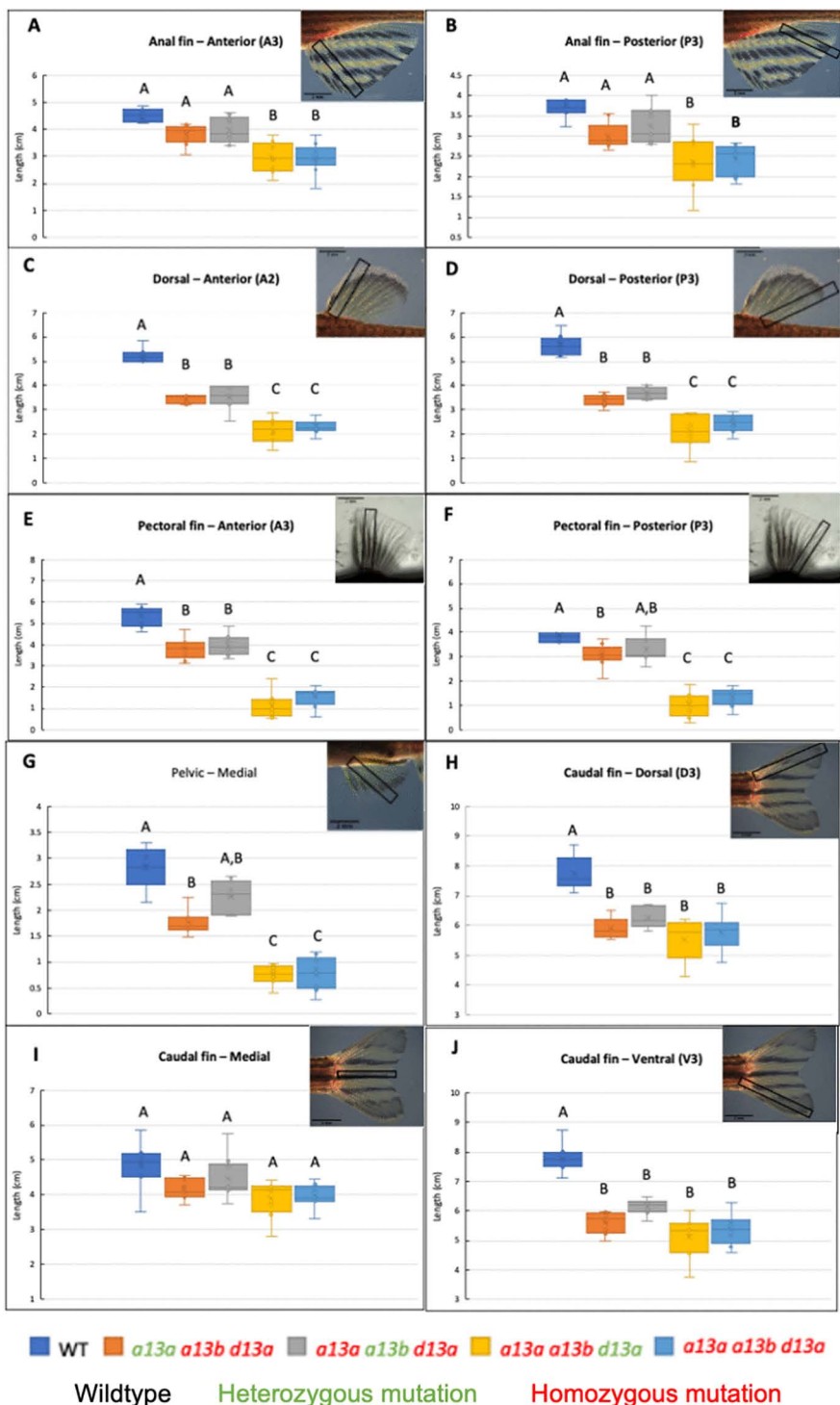

**Fig 2. Double and triple *hox13* mutants have reduced fin ray lengths in each fin compared to wildtype siblings. (A-J)** Box and whisker plots for ray measurements of selected rays in each fin. Wildtype siblings, double homozygous, and triple homozygous mutants are compared. The measured fin ray is shown at the top right corner of each corresponding graph. In the legend, green represents a heterozygous gene, and red represents a homozygous deletion. Means are presented for wildtype siblings (n=6), *hoxa13a⁻/⁺, hoxa13b⁻/⁻, hoxd13a⁻/⁻*, (n=9), *hoxa13a⁻/⁻, hoxa13b⁻/⁺, hoxd13a⁻/⁻* (n=11), *hoxa13a⁻/⁻, hoxa13b⁻/⁻, hoxd13a⁻/⁺* (n=8), and *hoxa13a⁻/⁻, hoxa13b⁻/⁻, hoxd13a⁻/⁻* (n=10) with maxima and minima represented by error bars. Significant groups were determined using 95% confidence intervals (p<0.05), and are represented as a, b, and c to show statistically significant differences from other groups.

## Varying contributions of *hoxa13a, hoxa13b,* and *hoxd13a* to fin ray defective phenotype

We next investigated the role of each *hox13* gene by examining adult fish (3-3.2 cm standard length (SL)) having only one wildtype allele for a given gene in the mutant genetic background for the other two genes. The most consistency between genotypes is observed in the caudal fin, where rays of all genotypes contain joints, bifurcations and actinotrichia (Figs 3, 4,

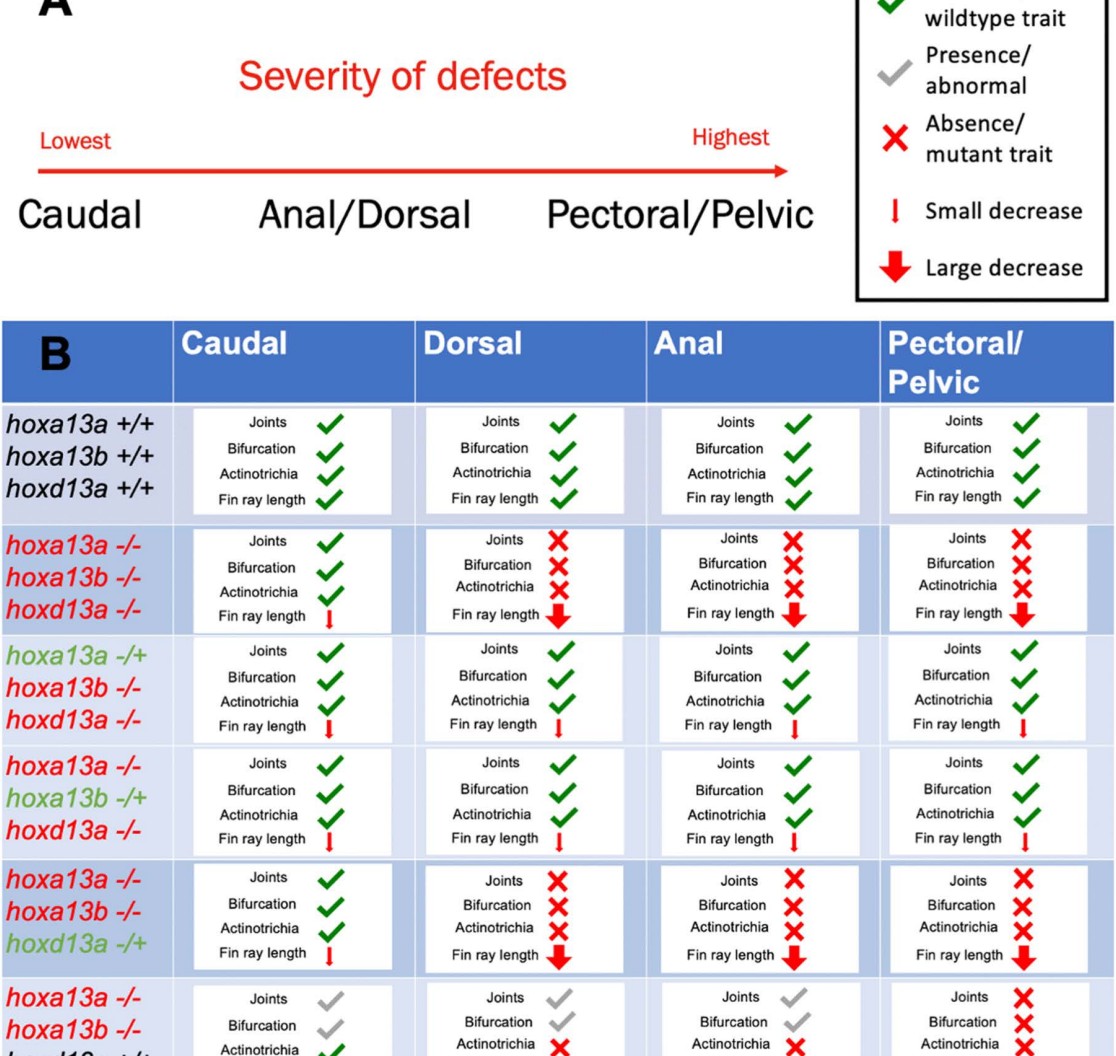

**Fig 3. Summary of defects in each fin type for triple and double homozygous *hox13* mutant fish. (A)** The caudal fin demonstrates the least severe defects in these mutants. The dorsal and anal fins lack joints, bifurcations and actinotrichia, but do not have length reductions as severe as the paired fins. The paired fins demonstrate the most severe defects in *hox13* double and triple homozygous mutants. **(B)** A summary of observed phenotypes in double and triple homozygous mutants for the presence/ absence of joints, bifurcations and actinotrichia, as well as severity of length reductions. The genotypes of the double and triple mutants are indicated in the left column: Heterozygotes for each gene are shown in green, homozygous mutants are shown in red, and wildtypes are shown in black. Number of fish observed for each group include n = 6 for wildtype siblings, n = 8 for *hoxa13a⁻/⁻, a13b⁻/⁻, d13a⁻/⁺*, and n = 10 for *hoxa13a⁻/⁻, a13b⁻/⁻, d13a⁻/⁻*, n = 11 for *hoxa13a⁻/⁻, a13b⁻/⁺, d13a⁻/⁻*, n = 9 for *hoxa13a⁻/⁺, a13b⁻/⁻, d13a⁻/⁻*, n = 2 for *hoxa13a⁻/⁻, a13b⁻/⁻, d13a⁺/⁺*. Bright field image examples for each mutant are shown in Fig 4.

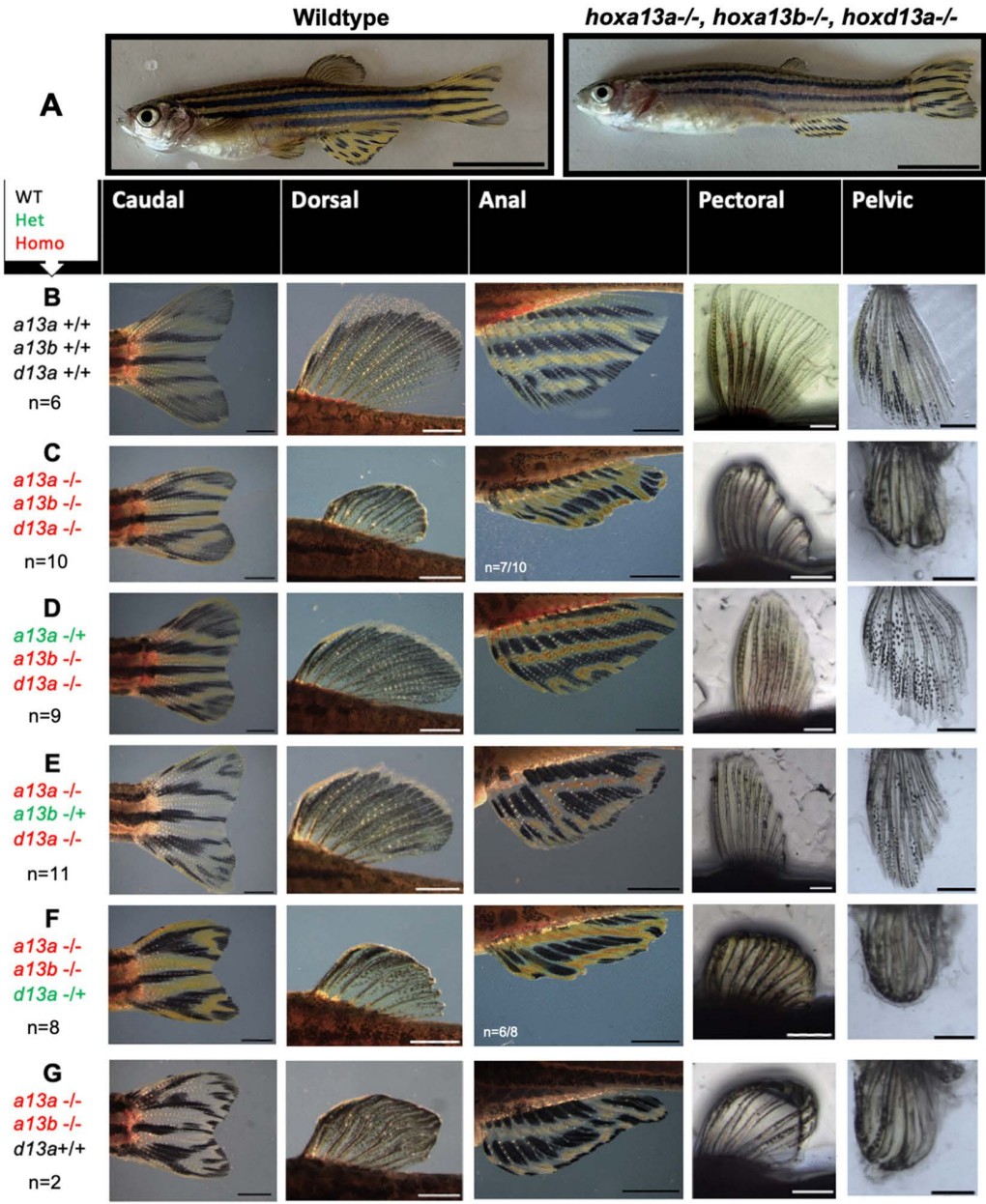

**Fig 4. Phenotypes of each fin in double and triple homozygous _hox13_ mutants. (A)** Whole-fish images of wildtype and triple _hox13_ mutant fish. Scale bars, 1 cm. **(B-G)** Caudal, dorsal, anal, pectoral and pelvic fin phenotypes for wildtype siblings, double homozygous, and triple homozygous mutants of interest. Replicates are listed as fractions for each genotype. The only inconsistency observed was joints appearing in the anal fins of mutants, as shown in S1C Fig. The genotypes of the fish are indicated on the left-hand side of the photos. Heterozygotes for each gene are shown in green, homozygous mutants are shown in red, and wildtypes are shown in black. Sample sizes are shown underneath the genotype of each fish. All fins of all genotypes showed high consistency. The only inconsistency was in the anal fin of the triple _hox13_ mutant, and the _hoxa13a_ -/-, _hoxa13b_ -/-, and _hoxd13a_ -/-, where sporadic, unorganized joints were observed in 3/10 and 2/8 replicates, respectively. Closeups images of these traits are also shown in Figs 1, S1, and S4. Scale bars, 2mm for caudal and anal fins, 1.5mm for dorsal fins, 1mm for pectoral fins, and 0.5mm for pelvic fins.

and S1). The only exception to this is the absence of actinotrichia at the tips of the lateral-most rays of the caudal fin of the *hoxa13a⁻ᐟ⁻, a13b⁻ᐟ⁻, d13a⁻ᐟ⁺* mutants and the triple homozygous mutants (S1D Fig). In addition to lacking actinotrichia, this lateral ray also develops much shorter than the other rays in the fin (S1D Fig), which does not typically occur in wildtype fins. The *hoxa13a⁻ᐟ⁻, a13b⁻ᐟ⁻, d13a⁻ᐟ⁺* mutant also has more inconsistent joint spacing compared to wildtype siblings, and reduced ray length in the lobes similar to what is observed in the triple *hox13* mutants (Fig 4F).

In all other fins (dorsal, anal, pectoral, and pelvic fins), defects differ depending on the genotype of the fish. In *hoxa13a⁻ᐟ⁺, a13b⁻ᐟ⁻, d13a⁻ᐟ⁻* mutants, and *hoxa13a⁻ᐟ⁻, a13b⁻ᐟ⁺, d13a⁻ᐟ⁻* mutants, regularly spaced joints, bifurcations and actinotrichia are present in all fin types similar to what is observed in wild type fish (Figs 3, 4D, 4E, S4A, and S4B). In *hoxa13a⁻ᐟ⁻, a13b⁻ᐟ⁻, d13a⁻ᐟ⁺* mutant however, joints, bifurcations, and actinotrichia are absent from all these fins similar to what is observed in the triple homozygous mutant (Figs 3, 4C, F, and S7). An exception to this however is observed in the anal fin, as joints can at times be observed in the rays of *hoxa13a⁻ᐟ⁻, a13b⁻ᐟ⁻, d13a⁻ᐟ⁺* mutants (n = 2/8) and the triple homozygous *hox13* mutant (n = 3/10) (S1C Fig). Adding one more *hoxd13a* allele (in the double mutants *hoxa13a⁻ᐟ⁻, a13b⁻ᐟ⁻, d13a⁺ᐟ⁺*) restores joints and bifurcations formation in both the dorsal and anal fin rays, although the joint spacing is very irregular (Figs 3 and 4G). However, joints and bifurcations are still missing in the paired fins of this mutant (Fig 4G).

These findings demonstrate that *hoxa13a* and *hoxa13b* play more critical roles in patterning the core structural features of soft rays in dorsal, anal, and paired fins, whereas *hoxd13a* contributes less to these processes. A single wildtype allele of *hoxa13a* or *hoxa13b* is sufficient to ensure proper formation of joints, bifurcation, and actinotrichia across all fin types. In contrast, *hoxd13a* requires two wildtype copies to partially rescue joint formation in dorsal and anal fins, albeit with irregular spacing and bifurcation patterns (Fig 4G). Single-copy *hoxd13a* alleles alone are insufficient to support these morphological features.

In addition, the length of the dorsal and anal fins of the triple *hox13* mutants along the anterior-posterior axis, when measured at their base, is also reduced (Fig 5A-5D and 5I-5J). Such reduced size is observed earlier in development as well, when the dorsal and anal fin primordia begin to appear in larvae at around 5.6mm SL (Fig 5E-H and 5K-5N). Gene expression analysis using *in situ* hybridization reveals that *hoxa13a, hoxa13b*, and *hoxd13a* are all expressed in the developing dorsal and anal fin primordia between 5.2-5.6mm SL (S5 Fig). Although there is a change in fin size, there does not, however, seem to be a change in the morphology, spacing, or number of vertebrae of these mutants. All triple *hox13* mutants possess 32 vertebrae, and with a size similar to the vertebrae of wildtype fish at the same SL.

### Phenotypic similarities between triple homozygous *hox13* mutant rays and acanthomorph spines

Zebrafish dorsal, anal, and paired fins typically consist of soft rays. However, in the triple homozygous *hox13* mutant, these fins exhibit rays resembling spiny rays of acanthomorph fish. To investigate this phenotype, skeletal analysis was performed using alizarin red staining on adult triple *hox13* mutants, their wildtype siblings, and *M. auratus*, an acanthomorph fish, for comparison. The triple *hox13* mutant rays share several characteristics with spiny rays: they do not bifurcate, lack joints, are relatively short, and appear thicker compared to wildtype soft rays (Fig 6). Based on these initial observations of the "spine-like" phenotype, further analysis was conducted to explore the relationship between true acanthomorph spines and *hox13* mutant rays.

Micro-CT analysis was performed on the dorsal fin rays of three triple *hox13* mutants and their wildtype siblings. Scans were reconstructed and reoriented in VGSTUDIO to obtain hemiray cross sections at standard positions along the proximal-distal axis of the ray (Figs 7A-C and S6B). The hemirays of the second (A2), fourth (A4), and sixth (A6) ray from the anterior side were analyzed at specific positions along the proximal-distal axis to compare the thickness of the bone between mutants and wildtype siblings (Fig 7A and 7B). The bone of the triple *hox13* mutant hemirays shows significant increases in thickness between 2 and 2.5 times thicker compared to wildtype at every position analyzed (Fig 7C and 7D). The most pronounced changes are seen in the A2 ray at the 500 μm and 750 μm positions, indicating a stronger thickness modification in the anterior, proximal part of the fin.

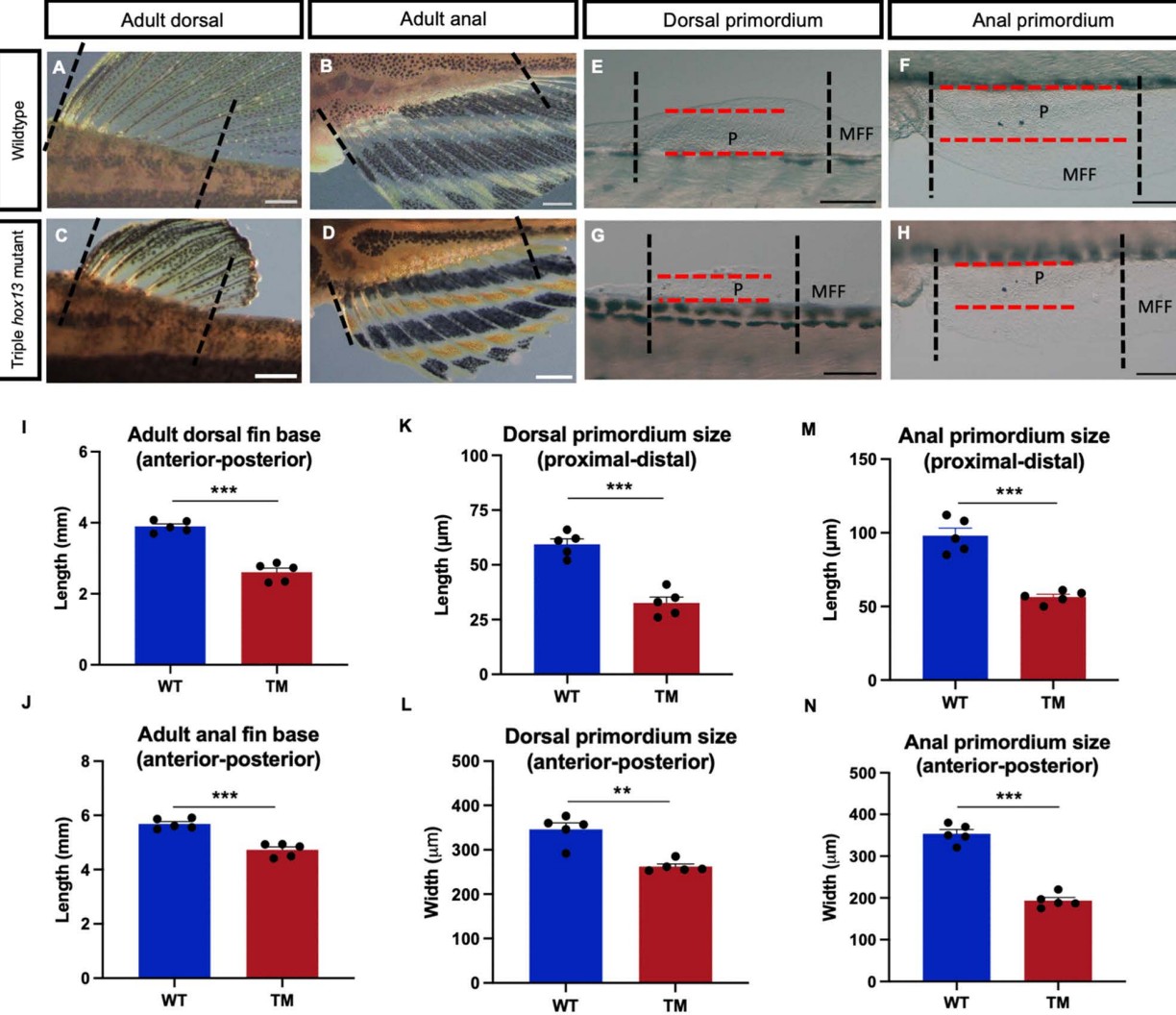

**Fig 5. Triple *hox13* mutants have proportionally smaller dorsal/anal fin primordia at 5.6mm SL, and smaller dorsal/anal fin base in adults.**
**(A-D)** Measurements of dorsal and anal fins at the base of the fin along the anterior-posterior axis of adult fish within 3-3.2 cm SL. Black dotted lines represent the distance measured. Scale bars, 1mm. **(E-H)** Measurement of length and width on wildtype dorsal and anal fin primordia at 5.6mm SL for wildtype and triple *hox13* mutants at 5.6mm SL. The black vertical dotted lines represent the anterior and posterior limits of the fin. The red dotted lines represent the proximal and distal limits of the fin primordia. The length and width of the fins were measured between these respective limits. P = primordium, MFF = median fin fold. Scale bars, 200 μm. **(I-J)** Average length (anterior-posterior) at the base of the dorsal and anal fin in 3-3.2 cm SL wildtype and triple *hox13* mutants. Measurements were averaged from n = 5 replicates for each group. Means + SE are presented. Unpaired t-tests were performed to test for statistically significant changes, which are indicated as * (p < 0.05), ** (p < 0.01), or *** (p < 0.001). **(K-N)** Average length and width of dorsal and anal fin primordia in 5.6mm SL wildtype and triple *hox13* mutants. Measurements were averaged from n = 5 replicates for each group at each location. Means + SE are presented. Unpaired t-tests were performed to test for statistically significant changes, which are indicated as * (p < 0.05), ** (p < 0.01), or *** (p < 0.001).

Despite the similarities mentioned above, the tips of the rays in triple mutants differ from that of the spiny rays of acanthomorphs. The two hemirays of the triple mutant remain separate, unlike the fused hemirays typically found in acanthomorph spines [10]. The dorsal and anal rays of triple mutants lack actinotrichia at their tips (Fig 1). This absence of actinotrichia is also evident in longitudinal and cross sections of fin regenerates (S7 Fig). Therefore,

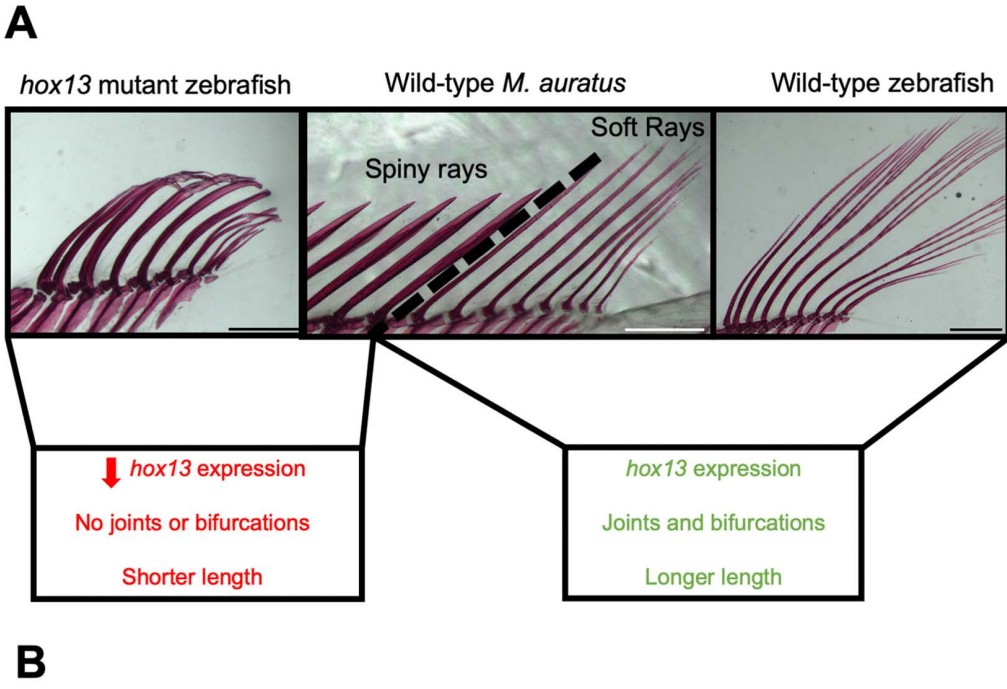

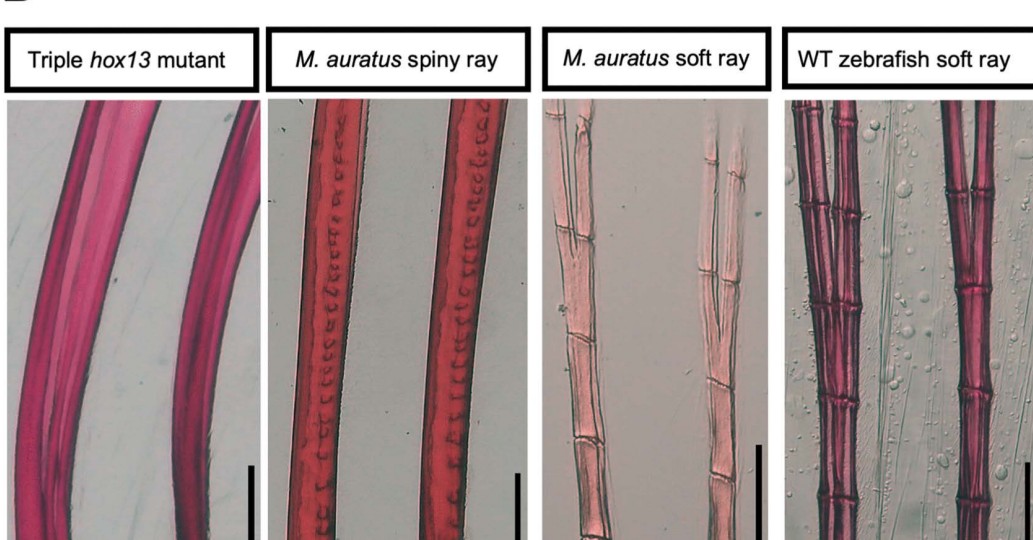

**Fig 6. Phenotypic similarities in the dorsal fin rays between the triple *hox13* mutants and acanthomorph spiny rays. (A)** Alizarin red staining of skeleton of the dorsal fins of triple *hox13* mutant, wildtype *M. auratus*, and wildtype zebrafish. Red text indicates the decrease or absence of a listed trait, while green text represents the presence or relative increase of a listed trait. Scale bars, 1mm. **(B)** Alizarin red staining on triple *hox13* mutant dorsal fin rays and *M. auratus* dorsal spiny rays. Two rays or spines are shown in each image. Triple mutant rays and spines are shown approximately halfway down the proximal-distal axis, and soft rays are shown at the first point of bifurcation. Scale bars, 200 μm.

instead of tapering at the tip as seen in WT soft rays, the bones of the triple mutants are thick and end abruptly beneath the fin epidermis. In contrast, the tips of spiny rays are sharp; they do not appear to contain actinotrichia between the partially fused hemirays, but instead have bundles of actinotrichia running along the posterior side of each ray (S7 Fig).

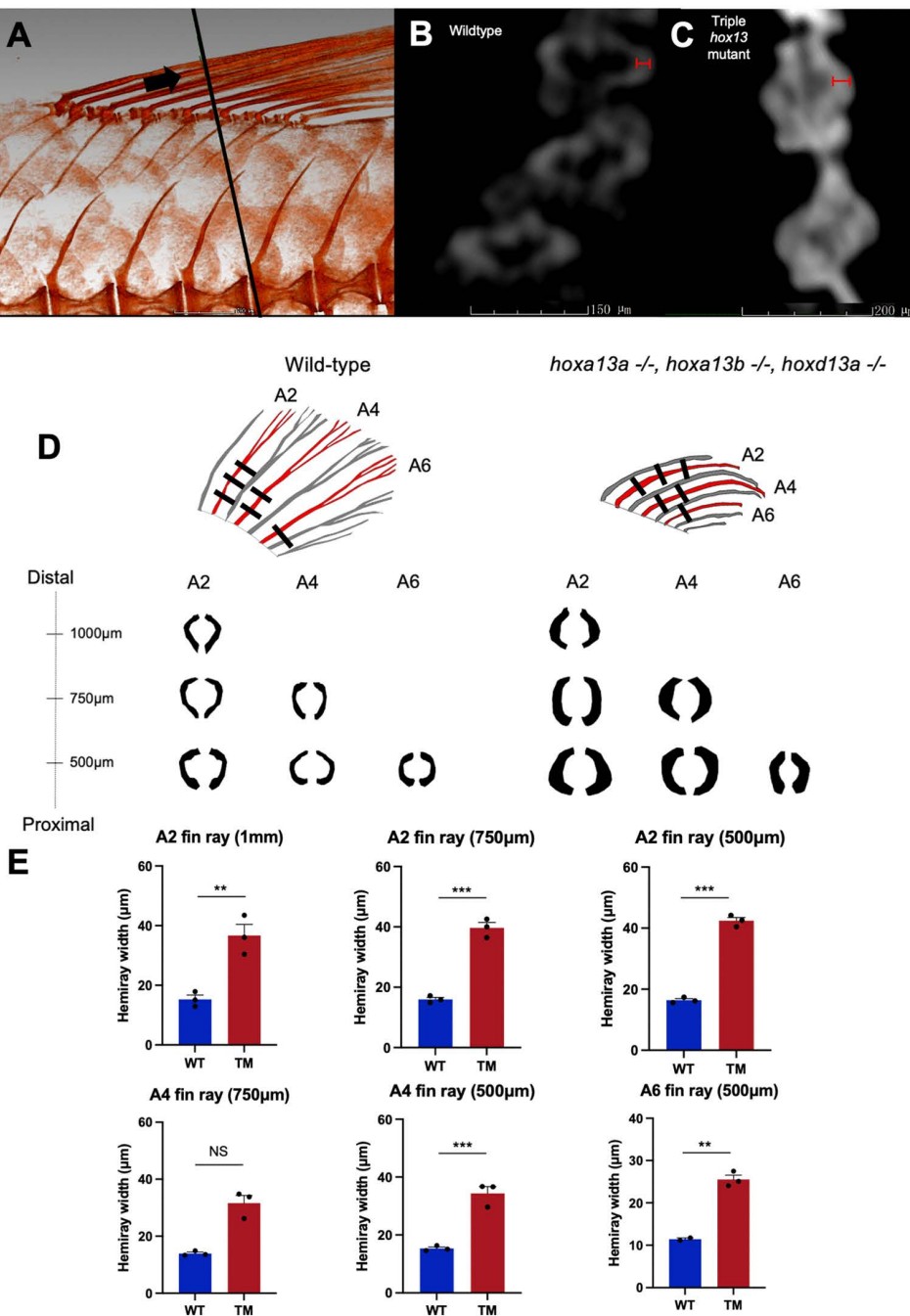

**Fig 7. Triple homozygous *hox13* mutants have thicker hemirays compared to wildtype siblings. (A)** Reconstructed wildtype zebrafish dorsal fin from micro-CT scan. Black line indicates the location of the transverse section through the ray shown in **(B)**. Scale bar, 700 μm. **(B)** Reconstructed cross section of wildtype dorsal fin hemirays. Red line indicates an example of measurements taken for **(E)**. Scale bar, 150 μm. **(C)** Reconstructed cross section of triple *hox13* mutant dorsal hemirays. Red lines indicated sample measurements taken for **(E)**. Scale bar, 200 μm. **(D)** Traces of wildtype and triple *hox13* mutant hemirays from reconstructed scans. Fin rays are named according to their position along the anterior-posterior axis (ex: A2 = second ray from the anterior side). Hemirays were observed at specific positions along the proximal distal axis shown on the left. These positions were selected for a consistent comparison between the wildtype and triple *hox13* mutant rays and demonstrate that the changes are consistent along the A-P axis. SL = 3.2 cm for both fish. **(E)** Bar graphs of dorsal hemiray widths in μm of wild-type and triple *hox13* mutants are presented at specific levels along the proximal distal axis. The ray of interest and distance from the base of the ray in μm is listed above each corresponding graph. Measurements were averaged from n = 3 replicates for each group at each location. Means + SE are presented. Unpaired t-tests were performed to test for statistically significant changes, which are indicated as * (p < 0.05), ** (p < 0.01), or *** (p < 0.001). A Mann-Whitney U test was performed for A4 fin ray (750 μm) since data was not normally distributed.

### *alx4a* and *grem1b* expression patterns differ in the triple *hox13* mutant dorsal and anal fins compared to wildtype siblings

As previously shown by Höch et al., [9] *grem1b* is normally expressed exclusively in the posterior part of the soft ray domain of acanthomorph fish before the emergence of the rays alongside *hoxa13a/b*. The *alx4a* gene is opposingly expressed exclusively in the anterior spiny ray domain of these fish and is confined to only the anterior fin rays in zebrafish [9,14]. Overexpression of *grem1b* in acanthomorphs reduces the *alx4a* domain of expression and results in a smaller spiny ray domain, while under-expression causes the opposite effect [9]. Given that our triple *hox13* mutants display spine-like rays, we investigated potential differences in *alx4a* and *grem1b* expression in these mutants compared to wildtype zebrafish prior to ray emergence in the dorsal and anal fins (around 5.4-5.6mm SL). In wildtype zebrafish, *alx4a* is exclusively expressed in the anterior portion of the fins (Figs 8A and S8A), whereas *grem1b* is exclusively expressed in the posterior part of the fin (Figs 8A and S8B). Considering the smaller size of the primordia of the triple mutants compared to the wildtype siblings, *alx4a* expression analysis reveals a relatively increased domain of expression in the triple *hox13* mutant dorsal and anal fin primordia (Figs 8A, S8A, and S8C-F). However, there is no difference of the overall *alx4a* expression level between genotypes (Fig 8B). In the case of *grem1b*, there appears to be a strong decrease in expression in the triple *hox13* mutant dorsal and anal fins (Figs 8A, 8B, and S8B).

Overall, the lack of expression of *hox13* in all fins except for the caudal fin has profound effects on the overall size of the fin and structure of the rays. In the dorsal and anal fins of triple *hox13* mutants, we also observe changes in *grem1b* expression, which plays an important role for establishing the soft ray domain in acanthomorph fish [9].

## Discussion

Morphological comparisons of double and triple homozygous compound mutant fish reveal varying roles of *hoxa13a, hoxa13b,* and *hoxd13a* in adult ray patterning in zebrafish. Both *hoxa13a* and *hoxa13b* prove to be more essential for typical soft ray formation in the dorsal, anal, and paired fins compared to *hoxd13a.* In contrast, mutations in these genes have a much milder impact on the caudal fin, causing only minor defects such as small length reductions and variable joint spacing. Interestingly, the dorsal, anal, and paired fins of the triple *hox13* mutant exhibit a phenotype more similar to acanthomorph spiny rays than wildtype zebrafish soft rays. These fin rays lack joints and bifurcations, have shorter length, and possess thicker hemirays. The molecular mechanisms causing these changes, however, do not seem to completely mirror what is observed in acanthomorph fish, as ubiquitous *alx4a* expression across the fin primordia was not observed in the triple *hox13* mutant.

From observations of double homozygous/ single heterozygous compound mutants, all three *hox13* genes of interest show some redundancy with one another in their role on ray joint and bifurcation formation. The joint defects observed in these mutants support our previous data suggesting *hoxa13a* involvement in initiation of joint formation during fin regeneration [32]. There is, however, a varying degree to which each of these genes affect the main ray characteristics. In the case of *hoxa13a/b*, only one copy of either gene is required for proper ray joint, bifurcation, and actinotrichia formation in all fins. In contrast, only one copy of *hoxd13a* is unable to exert this same effect. A difference in the baseline expression levels of the three *hox13* genes may be able to explain the respective effects of these gene mutations. From data mining of the RNA-seq analysis performed by Rabinowitz et al. [34] on the intact zebrafish caudal fin at different points along the proximal-distal axis, the expression levels of *hoxa13a*, *hoxa13b*, and *hoxd13a* differ greatly at the distal end. In the distal portion of the caudal fin, *hoxa13b* is showing the highest expression of the three genes with 1003.8 reads per million (RPM), *hoxa13a* expression is 279.9 RPM, and *hoxd13a* has the lowest expression level by far with only 14.6 RPM. Therefore, it is possible that all three *hox13* genes of interest have a similar function, but a difference in baseline expression level may explain the defects observed in the double homozygous *hox13* mutants. To support this further, the entire *hoxd* cluster in zebrafish can be deleted in zebrafish with little to no effect on body morphology [35], potentially indicating either a redundancy between this cluster and other *hox* clusters, or a lower importance of this cluster compared to others

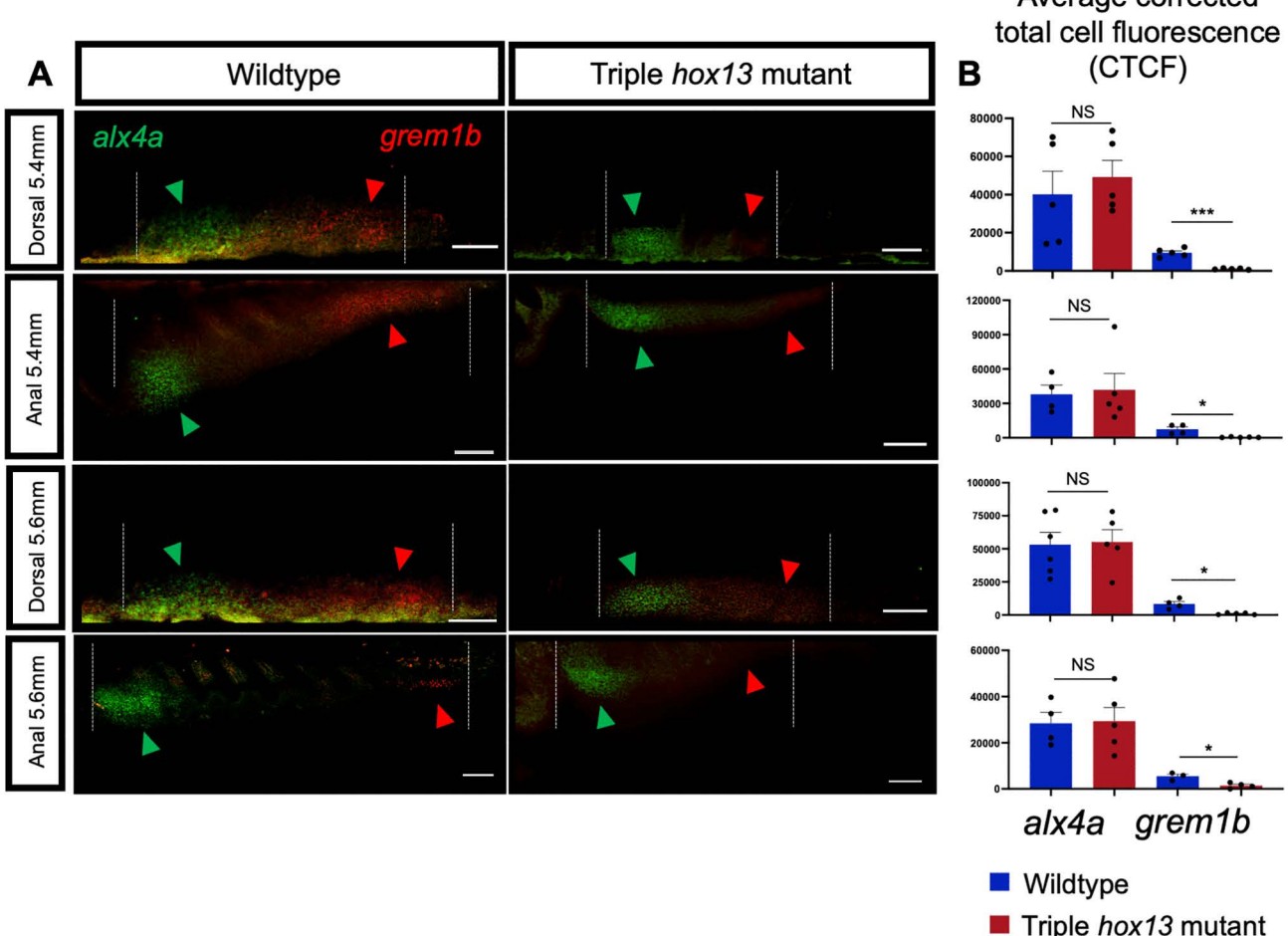

**Fig 8. Changes of expression patterns of *alx4a* and *grem1b* in triple *hox13* mutant larval zebrafish dorsal and anal primordia compared to wildtype. (A)** Gene expression analysis of *alx4a* and *grem1b* using HCR v3.0 RNA-FISH in wildtype and triple *hox13* mutant zebrafish at 5.4 and 5.6mm SL. *alx4a* expression is detected by the green fluorescence and *grem1b* expression is detected by the red fluorescence in the dorsal and anal fin primordia. Green and red arrowheads indicate the *alx4a* and grem1b domains of expression, respectively. White dotted lines indicate the anterior and posterior limits of each primordia. 5.4mm SL: n=5 for both genotypes. 5.6mm SL: n=4 for wildtype and n=5 for triple *hox13* mutants. Scale bars, 50 μm. **(B)** Average corrected total cell fluorescence (CTCF) for *alx4a* and *grem1b* in wildtype and triple *hox13* mutants for each fin and SL listed in **(A)**. CTCF graphs correspond to timepoints in adjacent panels in **(A)**. Means + SE are presented for each group. Unpaired t-tests were performed to test for statistically significant changes, which are indicated as * (p < 0.05), ** (p < 0.01), or *** (p < 0.001).

for morphology. Interestingly, acanthomorph fish completely lack *hoxd13a* [36], and are still able to form soft rays [9]. Certain species of herring including *Denticeps clupeoides* also lack both *hoxa13a,* and *hoxd13a* [36], while still being able to form soft rays in all fins, mirroring our results of only one of either *hoxa13a* or *hoxa13b* being necessary for proper soft ray formation.

A common theme observed in this study is that the caudal fin is showing very little variation between all mutant genotypes. Exactly why this occurs remains to be fully understood, however it could be a result of increased genetic compensation on this fin. From an evolutionary perspective, it is likely that the caudal fin has more compensatory mechanisms in place to conserve its structure, considering its importance for swimming. Amputations of the caudal fin have been shown to have significant effects on swimming speed and performance in many fish species including sockeye salmon [37], rainbow trout [38], goldish, carp, and qingbo [39]. Without the caudal fin, the thrust of the fish is extremely reduced

while swimming [40]. Considering this mechanistic importance, there may be more pressure to maintain a proper caudal fin since deformities or amputation in this fin may cause decreases in the fitness of the fish. Thus, more genetic compensatory mechanisms may be in place to maintain its structure. It could also be that there are simply different regulatory mechanisms or genes that are responsible for the development of the caudal fin when compared to the other fins. Some candidate genes for this caudal fin regulation include *hoxc13a* and *hoxc13b* that have been shown to be expressed in the embryonic caudal fin fold until 72hpf [41], however, it is not known if they are expressed as the rays are developing. Nevertheless, knockdowns of these genes after fin amputation led to significant reductions in proliferation and fin ray outgrowth suggesting a role for both *hoxc13* paralogs for caudal fin regeneration [42]. Recently, *hoxc13a* and *hoxb13a* have both been shown to be essential for the proper development of the caudal fin, as mutants with double homozygous deletions in these genes completely lack this fin [36]. The other fins however seem to develop normally in these double homozygous mutants [36]. Considering this finding, it is possible that the *hoxc13* and *hoxb13* genes are more important for caudal fin development, whereas the *hoxa13* and *hoxd13* genes are more important for the development of the other fins.

Beyond the rays of adult fins, *hox13* genes are also important for dorsal and anal fin primordia development. The anal and dorsal fin primordia are significantly smaller in the triple *hox13* mutants (Fig 5E-H and 5K-N). The overall fin size at the base of the dorsal and anal fins in the adult triple *hox13* mutants is also smaller compared to wildtype (Fig 5A-D, 5I, and 5J). To support this further, *hoxa13a, hoxa13b*, and *hoxd13a* were all found to be expressed in the dorsal and anal fin primordia between 5.2-5.6mm SL, showing that these genes likely have a role in the proper growth of these fins at early stages. In zebrafish, Nakamura et al. [31] reported a similar result for the pectoral fin folds of their *hoxa13a* [-/-], *hoxa13b* [-/-] mutants that show a 30% decrease in length (proximal-distal) at 72hpf, supporting the importance of *hox13* for the early fin growth. This similarity between the dorsal, anal, and pectoral fins also reinforces a probable similarity of developmental mechanisms between the paired fins and the dorsal and anal fins as suggested by Freitas et al [43]. In addition to zebrafish, developmental delays have been previously observed in *HoxD13* mutant mice, where phalangeal cell condensation and digit ossification were delayed in both forelimbs and hindlimbs of heterozygous mutants [44].

Spiny rays and soft rays have very different structures [9]. Morphologically, rays of the triple *hox13* mutant share some characteristics with spines, such as the lack of joints, bifurcations, shorter length, and thicker hemirays. Thus, the loss of *hox13* genes may be causing a partial transformation of the soft rays to form a structure more closely resembling spines.

The rays of the triple *hox13* mutants are also characterized by an absence of actinotrichia. Interestingly, it has recently been shown that certain species of acanthomorph, such as Rainbowfish (*Melanotaenia praecox*), have spiny rays that completely lack actinotrichia during development and during adulthood [45]. In contrast, adult *Melanochromis auratus* and *Pseudotropheus saulosi* that we used in the current study have actinotrichia fibers associated with the posterior part of their spiny ray tips. It is important to note that these actinotrichia are arranged differently compared to what is observed in soft rays (S7B Fig). Given this observation, the absence of actinotrichia or a different distribution of actinotrichia compared to soft rays may contribute to the diversification of spiny ray structures through variations in *hox13* gene expression.

The *hox* genes are essential for proper body patterning in vertebrates and organize positional information along the rostral-caudal axis of the body during development. Disruption of these *hox* genes have previously been associated with structural shifts in body structures. A common example of these changes includes homeotic shifts, which occurs via a genetic mutation causing one structure of the body to develop as another [46]. For example, mutations of the *Antp* gene in *Drosophila* can cause transformations of antenna into legs and vice versa [46,47]. In mice, a targeted disruption of *Hoxa11* causes a transformation of the thirteenth thoracic vertebrae to the first lumbar vertebrae [28]. In the limbs of chickens, ectopic expression of *Hoxd11* at the anterior side of the bud has been shown to cause a more posterior identity of the anterior digits [23]. Given that changes in structure are a common result of mutations to *hox* genes, we hypothesized that the difference in structure we observed in the triple *hox13* mutant rays could represent part of the transformation event from soft rays to spines in acanthomorphs. However, it is important to note that this possible transformation into spines is incomplete, as the anterior part of the hemirays are not fused, actinotrichia

are absent while spines still possess actinotrichia bundles extending from the posterior opening between the hemirays. This data however does remain speculative, and further analysis of multiple species with different fin ray variations and phylogenetic relationships would be needed to confirm this finding on a broader evolutionary scale.

In acanthomorphs, the spine domain in the developing dorsal and anal fins is not only characterized by an absence of *hoxa13a* and *hoxa13b* expression but also of *grem1b* [9]. Furthermore, gain and loss of function analyses have shown that *grem1b* is involved in the determination of the position of the spine to soft ray boundary [9]. In the zebrafish triple *hox13* mutants, *grem1b* expression is strongly reduced suggesting that formation of spine-like rays may result from this downregulation. This also suggests that *hox13* genes are acting as upstream regulators of *grem1b* in zebrafish dorsal and anal fins. It has previously been shown that deletions of the *HoxA* and *HoxD* clusters result in a complete lack of *Grem1* activation in the early mouse limb bud [48]. In addition, in this study, *Hoxa13* was also shown to upregulate a reporter line driven by a *Grem1* limb enhancer region in mouse limb bud [48]. This direct regulation is also supported by recent functional analysis showing the activity of *cis*-regulatory modules (CRMs) of mammalian *Grem1* enhancers containing multiple binding sites for HOX13 transcription factors in mouse limb buds [49]. Interestingly, the CRM2 enhancers were also found to be conserved in basal fishes and active in reporter transgenic mice limb bud, and mutagenesis of the HOX13 binding sites in CRM2 disrupted this activity [49]. On the other hand, *Hox13* genes are also known activators of *Shh* in the mouse limbs [50], which in turn, activates *Grem1* expression [51]. If similar regulation exists in zebrafish dorsal and anal fins, the absence of *hox13* expression in the triple *hox13* mutant fish may also potentially explain the downstream reduction in *grem1b* expression. As evidenced by our data, Hox13 factors in zebrafish likely activate *grem1b* through one of or both mechanisms.

On the other hand, although there is an expansion of *alx4a* domain of expression relative to the size of the dorsal and anal fins in the zebrafish *hox13* mutants, this expression does not encompass the entire developing fin primordia in the mutants. This observation contrasts with the situation in acanthomorphs where *alx4a* expression spans the spine domain. Such difference suggests that *alx4a* restriction to the anterior margin of the primordia is under the control of other factors than (or in addition to) the *hox13* genes in zebrafish. This may also potentially explain the partial transformation of the soft rays into spines in the zebrafish *hox13* triple mutants, as modification of other factors may be necessary to achieve a full *alx4a* expansion in these fins and result in a full transformation into spiny rays in zebrafish.

The spine-like phenotype of the triple *hox13* mutant rays suggest that the loss of *hox13* expression in soft rays (and consequent decrease in *grem1b* expression) may have been an important step towards the transformation of these rays to spines during the evolution of the acanthomorphs. Since the triple *hox13* mutants do not form true spines however, the loss of *hox13* is likely not the only factor involved in the evolution of this trait, suggesting that subsequent genetic modifications may have also been necessary.

In conclusion, the *hoxa13a, hoxa13b, and hoxd13a* genes have a variable yet undeniable importance for the proper patterning of the soft rays. A lack of expression of these genes in the zebrafish dorsal, anal, and paired fins causes fin rays to develop with a lack of joints, bifurcation, and actinotrichia, and with much shorter length. These changes in fin ray structure result in characteristics that resemble spiny rays of acanthomorph fish, suggesting that changes in *hox13* gene expression may have contributed to the appearance of spines during the evolution of acanthomorph fish.

## Methods

### Ethics statement

All experiments adhered to ethics policies for the University of Ottawa. Experimental procedures were reviewed and approved by the University of Ottawa Animal Care Committee. (approval number: BLe-3548).

### Animals

All zebrafish used for this project were bred and housed in the University of Ottawa aquatic facility. All fish are housed in a 28.5°C temperature-controlled room consisting of a 14- hour light/ 10-hour dark cycle. Fish were regularly fed by Animal Care and Veterinary Services (ACVS) staff at the university, and system water used in the tanks was UV sterilized. Animal

care and experiments were all performed according to Canadian Council on Animal Care (CCAC) guidelines. Three specimens of cichlid fish (one *Melanochromis auratus* and two *Pseudotropheus saulosi*) were obtained from a local pet store to use for comparisons with the zebrafish triple *hox13* mutant. Mutant fish containing deletions were obtained by successive crossing of fish with single mutations for *hoxa13a, hoxa13b,* and *hoxd13a*. Triple heterozygous crosses were used to establish a population of wildtype siblings to use for comparisons as well as the first adult triple *hox13* mutants.

## CRISPR/Cas9, micro-injection, and mutant screening

The target sites were selected using CRISPRscan [52] and sites that were used to generate point mutations in zebrafish *hox13* genes. For each *hox13* gene, two target sites were designed to achieve large fragment deletion as described before [53]. sgRNAs were synthesized using the cloning-free method as described [54]. Cas9 protein (EnGen Spy Cas9 NLS) was purchased from New England Biolabs. The final 10µl injection mix was made with 10–20ng/µl sgRNAs and 320ng/µl Cas9 protein. Primary injected fish were bred with wildtype for PCR screening as described [55]. The resulting mutations were finally verified by sequencing (S1B Fig).

## Genotyping

Scales were removed from zebrafish according to the protocol outlined in Rasmussen et al. [56] and boiled in 50mM NaOH for 10 minutes to extract DNA. Fish were genotyped using the following sets of primers: *hoxa13a* F: 5'-AGG CGAAGATTATACCAGCTCAC-3', *hoxa13a* R1: 5'-CATCAAACAACTCATCCTTTGG-3', *hoxa13a* R2: 5'CCTGTC-GTTCAGATAGGTTGG3', *hoxa13b* F: 5'-CCACCACTTTGTTTCAGTTCAA-3', *hoxa13b* R1: 5'-TGATGCCCTTGT ACTTGTTGAC-3', *hoxa13b* R2: 5'-ATATCCATAGGGCAAAGAAGCA-3', *hoxd13a* F: 5'-TTTACCCATCTGCCTTCGGG-3', *hoxd13a* R1: 5'AGACCTCTTGCAGTCAAGGT3', *hoxd13a* R2: 5'-CTGCTGCAATTGTTTGACCAGT-3'. For each gene, the R1 primer is located at the end of the deletion, and the R2 primer is located within the deletion. Expected band sizes are listed in S1 Table, and an example of an agarose gel is shown in S1A Fig.

## Live fish imaging

Experimental fish were anesthetized in a tricaine solution at a concentration of 0.2mg/mL. Fish were placed under a Leica MZFLIII dissecting microscope on a petri dish containing a 2% agarose gel to be used as a background. Images were captured using a Luminera Infinity 3 camera to observe the presence or absence of joints, bifurcations, and actinotrichia in *hox13* compound mutants. Close up images of rays and actinotrichia were taken on a Axiozoom V16 dissecting microscope with an Axiocam 208 color camera. Image measurements and analysis was performed in ImageJ.

## Bone staining

Bone staining was performed according to the protocol previously described [57] with minor modifications. Samples were incubated in B-staining solution for 1 hour at room temperature with gentle rocking.

## Probe synthesis and *in situ* hybridization

Antisense riboprobes for *alx4a* and *grem1b* were generated from 48hpf WT cDNA using PCR as previously described [58]. Partial sequences of the coding sequences were used for each. The sequence to make the *alx4a* probe was amplified using the previously described primers [59]: *alx4a* F: 5' ATGAACGCCGAGACGTGC 3'; *alx4a* R: 5' TCATGTAGCCCAAGAGATGGC 3' (1096 bp amplicon) and were amplified again using the primers: T7 *alx4a* F: 5' CAGTGAATTGTATACGACTCACTATAG GGAGATCATGAACGCCGAGACGTGC 3' and Sp6 *alx4a* R: 5' CAGTGAATTGATTTAGGTGACACTATAGAAGTCTCATG-TAGCCCAAGAGATGGC 3'. The sequence to make the *grem1b* probe was amplified using the primers: *grem1b* F: 5' TGCAC-GGTGACAGATTCTGC 3', and *grem1b* R: 5' GGACGCTTCACAGATCGTTTC 3' (849 bp amplicon) and were amplified again

using the primers: T7 *grem1b* F: 5' CAGTGAATTGTATACGACTCACTATAGGGAGATGCACGGTGACAGATTCTGC 3' and Sp6 *grem1b* R: 5' CAGTGAATTGATTTAGGTGACACTATAGAAGTCGGACGCTTCACAGATCGTTTC 3'. The antisense RNA probe for *alx4a* was then transcribed *in vitro* using SP6 RNA polymerase, an antisense probe for *grem1b* was transcribed using T7 RNA polymerase. *Hoxa13a* (500 bp) and *hoxa13b* (700 bp) antisense RNA probes were generated as previously described [60]. *Hoxd13a* cDNA was cloned into pDRIVE using the primers: F: TGGGATTAACATTTGATGCAGACG and R: CCCATG-CACTGAGGAATATGGAC. The antisense RNA probe was synthesized using SP6 RNA polymerase. Larval zebrafish were fixed at 4°C overnight in 4% paraformaldehyde and stored in 100% MeOH at -20°C until use. Whole mount *in situ* hybridization was performed on 5.2-5.6mm SL zebrafish larvae using the protocol described in [61], using a 12-minute proteinase K digestion at 10µg/mL. Stained fish were then imaged on an Axiozoom V16 dissecting microscope.

### Hybridization chain reaction (HCR) RNA-FISH

For target mRNAs, a kit containing a DNA probe set, HCR amplifiers, as well as hybridization and amplification wash buffers was obtained from Molecular Instruments (molecularinstruments.com). Probe sets for *alx4a* were used with an X1 hairpin amplifier system labelled with a 488nm fluorophore, and probe sets for *grem1b* were used with an X3 hairpin amplifier system labelled with a 647nm fluorophore. Larval zebrafish were fixed at at 4°C overnight in 4% paraformaldehyde and stored in 100% MeOH at -20°C until use. HCR v3.0 FISH was performed according to protocols listed on molecularinstruments.com and from Choi et al. [62]. Samples were mounted in Vectashield plus antifade mounting medium, imaged on an Olympus FV1000 BX61 LSM Confocal microscope. Using ImageJ, images were processed and corrected total cell fluorescence (CTCF) was measured using the formula: CTCF = Integrated Density – (Area × Mean Background Fluorescence) according to McCloy et al. [63].

### Micro-CT scanning

Three samples for both the triple *hox13* mutant, and wild-type siblings were Micro-CT scanned using a SkyScan1173 Micro-CT scanner (Software v. 1.6). Samples were first stained in 1% iodine solution in 99% EtOH overnight. Samples were then wrapped in paper towel and loaded into the scanner in a plastic tube facing vertically. They were scanned with the x-ray source set at a voltage of 60kV and 133µA. Voxel size was set at 7.1µm for each scan, with the object to source distance at 52.14mm, and the camera to source distance at 364mm. Scans were performed with an exposure of 1884ms and a rotation step of 0.35 degrees. Reconstruction of slices and analysis was performed using VGSTUDIO MAX software. Scans were oriented to obtain cross sectional images of the fin rays at specific positions along the proximal distal axis as represented by the schematic in Fig 6. Specific locations along the proximal distal axis of fin rays were chosen that allowed for consistent comparisons between the triple *hox13* mutant and wildtype hemirays. Locations were chosen to be more proximal within the fin ray as the distal parts of the rays are very small, making reconstructions more difficult to interpret. A sample of a fully reconstructed scan and raw images of fin ray cross sections used for measurements is shown in S6 Fig.

### Statistical analyses

Statistics and graphs were performed and generated using SPSS Statistics and Microsoft Excel. The normality of groups was tested using Shapiro-Wilks tests. Two-tailed Welch's T tests were used to assess significance between normally distributed groups. A Mann-Whitney U test was performed to assess significance of the non-normally distributed group (S8 Fig). To compare groups in Fig 2, 95% confidence intervals for each group were determined using descriptive statistics and compared. Graphing was performed in Microsoft Excel and Graphpad Prism 10.

### Supporting information

**S1 Table. Primer and sgRNA sequences used for genotyping, probe amplification, and CRISPR-Cas9 experiments.**
(XLSX)

**S1 Fig. Additional phenotypes observed in the triple *hox13* mutant fins. (A)** Example of agarose gel for genotyping triple heterozygous *hox13* mutant (*hoxa13a* <sup>-/+</sup>, *hoxa13b* <sup>-/+</sup>, *hoxd13a* <sup>-/+</sup>). Reverse primers were used with corresponding forward primers to generate band sizes outlined in S1 Table. Two individual reactions are performed for each gene. For reactions with the R1 primers, both primers span the deletion, meaning smaller bands correspond to the DNA containing a deletion, and a larger band corresponding to the wildtype sequence for a given gene. For reactions with the R2 primers, this primer is located within the deletion and will give a band for wildtype fish but will give no band for fish with homozygous deletions. **(B)** Sequences of *hoxa13a, hoxa13b,* and *hoxd13a* deletions in the deletion mutants compared to wildtype. The ends of the deletions are shown within the sequence for each gene. **(C)** Minority phenotype of triple *hox13* mutant dorsal fin, with joints present in the anal fin. White arrowheads indicate joints in the fin rays. Only n = 3/10 mutants in this analysis displayed this phenotype. Scale bars, 2mm. **(D)** Tips of the lateral rays of triple *hox13* mutant caudal fin. The orange arrowhead shows the tip of the lateral most rays, which lacks actinotrichia in the triple *hox13* mutant. The black arrowheads show fin rays with normal actinotrichia at the tip. Scale bar, 200um. **(E)** Closeup fin ray images of *hoxa13a* <sup>-/-</sup>, *hoxa13b* <sup>-/-</sup>, *hoxd13a* <sup>-/+</sup> mutants display a very similar phenotype to triple *hox13* mutants. Caudal, anal, dorsal, pectoral and pelvic fins are presented. White arrows represent joints which form in the caudal fin and can at times form in the anal fin of this mutant. Bifurcations are indicated with yellow arrowheads, which only occur in the caudal fin of this mutant. In contrast to the caudal fin, the tips of the other rays lack actinotrichia, which are indicated by orange arrowheads. Scale bars, 200 μm. (F) The urogenital pore of the triple *hox13* mutant is smaller compared to wildtype siblings. The outer structure of the pore is outlined with a black dotted line for each genotype. Scale bars, 1mm. (G) Length measurements for wildtype and triple *hox13* mutant urogenital pore along the proximal-distal axis. Means + SE are presented for each. An unpaired t-test was performed to assess statistical significance (\*\*\* = p < 0.001).
(TIF)

**S2 Fig. Numbers of fin skeletal elements of the triple *hox13* mutants compared to the wildtype siblings.** (A-L) Alizarin red staining of the fin rays and endoskeleton for each fin of the triple *hox13* mutants and wildtype siblings. (A-B) There is an increase in the number of distal radials in the triple mutants. Pectoral distal radials are numbered 1–7 in the wildtype (n = 5/5), and 1–10 in the triple *hox13* mutant (n = 4/4). Scale bars, 1mm. (C-D) The triple *hox13* mutants have the same number of endoskeletal elements as the wildtype siblings. Distal radials are numbered 1–3 in triple *hox13* mutants (n = 4/4) and wildtype siblings (n = 5/5). Scale bars, 1mm. (E-F) Decreased numbers of fin rays in the triple *hox13* mutant paired fins. n = 10/12 mutants have 9 rays in their pectoral fin compared to 10 in the wildtype and n = 6/6 mutants have 5 rays instead of 7 rays in wildtype (n = 5/5) for the pelvic fin. Scale bars, 1mm. (G-H) Dorsal fin rays and endoskeleton of triple *hox13* mutants and wildtype siblings: 7/12 mutants have 7 rays and endoskeletal elements compared to 8 in the wildtype (n = 5/5). P = proccurent ray, Pr = proximal radial. Scale bars, 1.5mm. (I-J) Anal fin rays and endoskeleton of triple *hox13* mutants and wildtype siblings: 6/12 mutants have 12 rays and endoskeletal elements compared to 13 in the wildtype (n = 5/5). P = proccurent ray, Pr = proximal radial Scale bars, 1.5mm. (K-L) There is no change in the number of endoskeletal elements and rays in the caudal fins of triple *hox13* mutants (n = 4/4) and wildtype siblings (n = 5/5). Hyp = hypural, phyp = parahypural, hs = haemal spine, pu = preural. Scale bars, 2mm.
(TIF)

**S3 Fig. Numbers of fin rays and distal radials differ in triple *hox13* mutant fins.** (A-B) Number of fin rays (A) and distal radials (B) in wildtype and triple *hox13* mutant fins for the pectoral, pelvic, dorsal, and anal fins. Each point represents a single fish, with wildtype individuals represented as blue dots, and triple mutants represented as red triangles.
(TIF)

**S4 Fig. *hoxa13a* <sup>-/+</sup>, *hoxa13b* <sup>-/-</sup>, *hoxd13a* <sup>-/-</sup> and *hoxa13a* <sup>-/-</sup>, *hoxa13b* <sup>-/+</sup>, *hoxd13a* <sup>-/-</sup> mutants present similar fin characteristics to wildtype fish.** (A-B) Closeups fin ray images of *hoxa13a* <sup>-/+</sup>, *hoxa13b* <sup>-/-</sup>, *hoxd13a* <sup>-/-</sup> and *hoxa13a* <sup>-/-</sup>, *hoxa13b* <sup>-/+</sup>, *hoxd13a* <sup>-/-</sup> fish, which resemble wildtype fin phenotypes with joints and bifurcations present in each fin.
(TIF)

**S5 Fig. *hox13* genes are expressed during early development of the dorsal and anal fin primordia.** (A-C) *In situ* hybridization using an antisense RNA probe for *hoxa13a, hoxa13b,* and *hoxd13a* in the dorsal and anal fin primordia of 5.2, 5.4, and 5.6mm SL wildtype fish. Scale bars, 100 μm for dorsal fins and 150 μm for anal fins. (TIF)

**S6 Fig. Sample CT reconstructions and fin ray cross sections of wildtype and triple *hox13* mutant fish.** (A) Whole-scan reconstructions of wildtype and triple *hox13* mutant fish (B) Dorsal fin ray cross sections at the same proximal-distal location in wildtype and triple *hox13* mutants. (TIF)

**S7 Fig. Actinotrichia are absent from triple *hox13* mutant dorsal and anal fin rays.** (A) Picrosirius red staining of a longitudinal lepidotrichia section of 5-day post-amputation (dpa) regenerates for wildtype and triple *hox13* mutant caudal and anal fins. Black arrowheads indicate actinotrichia within the regenerate, blue arrowheads indicate bone, and green arrowheads indicate epidermal tissue in each section. Scale bars, 100 μm. (B) Brightfield image of the tip of the intact dorsal fin of triple *hox13* mutant and wildtype zebrafish, as well as *M. auratus* spiny and soft rays. Black arrowheads indicate the presence of actinotrichia. Scale bars, 50 μm for soft rays and triple *hox13* mutant rays, 100 μm for spines. (C) Picrosirius red staining of cross sections of wildtype and triple *hox13* mutant anal fin regenerate at 5dpa and of intact dorsal spine of M. auratus. Cross sections in zebrafish rays (A&B) are at approximately the same distance from the distal tip as in the M. *auratus* cross section (C). Black arrowheads indicate actinotrichia within the regenerate, blue arrowheads indicate bone, and green arrowheads indicate epidermal tissue in each section. Scale bars, 50 μm. (TIF)

**S8 Fig. Expression of anterior-posterior patterning genes in wildtype and triple *hox13* mutant zebrafish.** (A) Gene expression analysis using *in situ* hybridization with for *alx4a* at 5.4-5.6mm SL in the dorsal and anal fin primordia for wildtype and triple *hox13* mutants. 5.4mm SL: n = 4 for each genotype; 5.6mm SL: n = 5 for each genotype. P = primordium, MFF = median fin fold. Black arrowheads point to the domain of expression. Scale bars, 100 μm for dorsal fins and 150 μm for anal fins. (B) Gene expression analysis of *grem1b* using *in situ* hybridization at 5.4mm SL in the dorsal and anal fin primordia in wildtype and triple *hox13* mutant. 5.4mm SL: n = 6 for wildtype, n = 4 for triple *hox13* mutants; 5.6mm SL: n = 4 for wildtype, n = 3 for triple *hox13* mutant. P = primordium, MFF = median fin fold. Black arrowheads point to expression. Scale bars, 100 μm for dorsal fins, 150 μm for anal fins. (C-D) Average length of the *alx4a* expression domain along the anterior-posterior axis in the dorsal and anal fin primordia of wildtype and triple *hox13* mutants at 5.6mm SL. Measurements were pooled from n = 5 replicates for each genotype. Means + SE are presented for each group. (E-F) Average size of the *alx4a* expression domain along the length anterior-posterior axis relative to the size of the anal or dorsal primordium in 5.6mm SL wildtype and triple *hox13* mutants. The size of the *alx4a* expression domains was divided by the size of the fin primordium and averaged. n = 5 replicates used for each calculation. Means + SE are presented for each group. (TIF)

## Acknowledgments

We acknowledge Simon Monis, Negar Arasteh, and Matthew Da Costa for their technical contributions and Dr. Fred Gaidies and the Carleton U. XR-CT laboratory for technical assistance on the CT scans. We thank Dr. Marc Ekker for discussion and critical reading of the manuscript.

## Author contributions

**Conceptualization:** Marie-Andrée Akimenko.

**Data curation:** Jordan Corcoran, Hailey Quigley, Qingming Qu.

**Formal analysis:** Jordan Corcoran.

**Funding acquisition:** Marie-Andrée Akimenko.

**Investigation:** Jordan Corcoran, Hailey Quigley.

**Methodology:** Jordan Corcoran, Hailey Quigley, Qingming Qu, Marie-Andrée Akimenko.

**Project administration:** Marie-Andrée Akimenko.

**Resources:** Qingming Qu.

**Supervision:** Marie-Andrée Akimenko.

**Writing – original draft:** Jordan Corcoran.

**Writing – review & editing:** Jordan Corcoran, Hailey Quigley, Qingming Qu, Marie-Andrée Akimenko.

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
