## [Decision Letter · Decision Letter 0]

17 Sep 2025

PGENETICS-D-25-00736

Combined mutations of hoxa13a, hoxa13b, and hoxd13a lead to structural shifts in zebrafish soft fin rays providing insight into spiny ray evolution

PLOS Genetics

Dear Dr. Akimenko,

Thank you for submitting your manuscript to PLOS Genetics. All reviewers were positive about the overall results of your manuscript. After careful consideration, we feel that it has merit but the reviewers comments should be addressed prior to publication in PLOS Genetics. Therefore, we invite you to submit a revised version of the manuscript that addresses the points raised during the review process.

Please submit your revised manuscript within 60 days. If you will need more time than this to complete your revisions, please reply to this message or contact the journal office at plosgenetics@plos.org. Please include the following items when submitting your revised manuscript:

We also apologize for the time this took to review. We had a reviewer who became unresponsive. We look forward to receiving your revised manuscript.

Kind regards,

Jeffrey J Essner

Academic Editor

PLOS Genetics

Pablo Wappner

Section Editor

PLOS Genetics

Aimée Dudley

Editor-in-Chief

PLOS Genetics

Anne Goriely

Editor-in-Chief

PLOS Genetics

**Additional Editor Comments:**

Reviewer #1:

Reviewer #2:

Reviewer #3:

**Journal Requirements:**

2) We noticed that you used the phrase 'data not shown' in the manuscript. We do not allow these references, as the PLOS data access policy requires that all data be either published with the manuscript or made available in a publicly accessible database. Please amend the supplementary material to include the referenced data or remove the references.

- ® on page: 31.

5) We notice that your supplementary Figures, and Tables are included in the manuscript file. Please remove them and upload them with the file type 'Supporting Information'. Please ensure that each Supporting Information file has a legend listed in the manuscript after the references list.

Potential Copyright Issues:

i) Please confirm (a) that you are the photographer of 1B, 2, 3A-H, 4A, 5A, 6A, S1C-E, S2, S3, S5, and S6A, or (b) provide written permission from the photographer to publish the photo(s) under our CC BY 4.0 license.

**Reviewers' comments:**

Reviewer's Responses to Questions

**Comments to the Authors:**

Reviewer #1: I have reviewed the manuscript entitled "Combined mutations of hoxa13a, hoxa13b, and hoxd13a lead to structural shifts in zebrafish soft fin rays providing insight into spiny ray evolution" by Jordan Corcoran1, Hailey Quigley1, Qingming Qu1,2, Marie-Andrée Akimenko1 with great interest. The paper is interesting and relevant as it demonstrates expression and functional data defining the role of three Hox13 genes in the development and specification of fin rays and spines. Because a great deal of morphological disparity in vertebrates occurs in fins, yet underlying genetic mechanisms specifying fin ray and spine morphology and identity are lacking, I felt this paper should be published. I had very few comments on the manuscript, which was well written and clearly laid out. I have uploaded a pdf with my comments.

Reviewer #2: Overall, the authors present an interesting study on the role of Hox13 paralogues during fin ray formation in zebrafish. They show evidence that different Hox13 combinatorial mutants exhibit defects in ray length, joint formation, bifurcation, and actinotrichia formation, phenocopying aspects of spiny ray formation. Collectively, these data support a role for Hox13 genes in proper soft ray development and raise interesting questions around the molecular mechanisms underlying the evolution of spiny ray morphologies in nature. As part of this work the authors generated new stable lines of triple mutants for hoxa13a -/-, hoxa13b -/-, hoxd13a -/-, which they used throughout their analyses. Overall, the study provides detailed assessments of mutant ray phenotypes. However, there are several issues that need to be resolved.

The phrase “typically have” is used in the results to describe the complement of rays or radials present in a particular strain. However, quantifications are not always presented.

Line 156: Avoid using phrases like “…see next section” within the body of the text. It is generally disruptive to the flow of the narrative.

Supplementary Figure S3. Apologies if I missed it, but the legend does not define the “A”, “B”, and “C” above the bar graphs. Additionally, can the genotypes be added directly to the graphs? This may be easier to read than the different color-codings for alleles and genotypes.

Lines 158, 190. The author refers to “inconsistency in joint spacing” in mutants compared to wild types. Has this been quantified?

Line 197: The authors state “…joints, bifurcations, and actinotrichia are absent from all these fins…” and refer to

Figure 2C/F. However, I found these details to be hard to see in the images presented. Perhaps adding high magnification insets might make the anatomical defects easier to visualize?

Line 198: The author states regarding the anal fin “…as joints can at times be observed in the rays of hoxa13a -/-, a13b -/- d13a -/+ mutants (n=2/8) and the triple homozygous hox13 mutant (n=3/10)” and refer to Figure S1A. However, this panel is a PCR gel and is incongruent with the text.

Line 270: A bracket is added as a typo at the end of the sentence.

In Figure 3, the authors show representative dorsal and anal fins for SL5.6mm larvae. However, according to the legend, the quantifications presented in I-L are for 3-3.2mm larvae. It is unclear why different fish were used. Also, given the variability in length of the fish, any comparisons would be strengthened by normalizing the length of the fin primordia by the length of the fish.

Figure 5 should have a panel added that shows a representative cross sectional MicroCT rendering of mutant rays in addition to the WT rays presented. Also, I would advise to remove the cartoon eye from Figure 5C.

The authors examine hemi-rays of the second (A2), fourth (A4), and sixth (A6) ray from the anterior side. Can the authors elaborate on why the specific rays and positions along the proximal-distal axis of the rays were selected for comparison, and why different numbers of sites were used between individual rays?

The authors measure the length of the expression domain of Alx4 in WT versus triple mutant fish. Apologies if I missed it, but the results text does not refer to panels C-F in Figure 6. In Figure 6E,F, if I understood correctly, the length of the expression domain was normalized to fin length. If so, the units should cancel and not be included in the Y axis label which is a ratio of the lengths. The authors claim that there is a decrease in grem1b expression in the triple hox13 mutant dorsal and anal fins. Is this based on staining intensity, or a type of quantification?

Line 441 and 459: Avoid phrases like “As previously explained” or “As previously mentioned”.

Given that triple Hox13 mutant rays phenocopy spiny rays, the authors argue changes in Hox13 expression may provide a mechanistic explanation for the evolution of spiny rays in nature. The discussion would benefit from the authors further elaborating on this model.

There are grammatical errors and typos within the text that need to be corrected.

Reviewer #3: This manuscript addresses the tole of a set of Hox genes in zebrafish fins. The experimental design incorporates a range of technical approaches including rather complex genetics, morphological analysis, Micro-CT scanning and reconstruction, and gene expression analysis. The work makes use of triple mutants (Hox a13a, a13b, d13a), which are homozygous viable but sterile and must be regenerated with crosses for each analysis. In addition, the role of individual genes is tested by “rescue” with one wild type copy of each of the genes. The phenotypes associated with these genotypes are carefully analyzed and demonstrate effect on all zebrafish fins, with general shortening of rays that are thicker than wild type and display loss of joints, bifurcations and actinotrichia. The authors carefully characterize these phenotypes in the different mutant backgrounds, revealing differential effects on different fins and differential contributions of the different Hox genes to the phenotypes, which is all nicely summarized in Table 1. The analysis leads to the speculation that Hox 13 genes were involved in the evolutionary transformation of soft rays to spiny rays.

This is a well carried out, carefully designed study, and the analysis appears to be quite rigorous. There is keen attention paid to details and the conclusions follow logically from the results. Although this work is highly specialized, I found it clearly written and appropriate for a broad scientific audience. Below I suggest revisions to improve the manuscript:

General comments

1. There is reasonably large variation in phenotype between individuals and even between fins of the same individual. This requires some explanation, especially since the number of animals examined is relatively small.

2. The evolutionary hypothesis is quite interesting and is consistent with the data. However, it is still speculative and correlative at this point in time. The wording should be softened throughout to make this clear. Experiments would need to be done examining multiple species with different phylogenetic relationships and trait variations in order to make stronger statements.

3. The in situ hybridization results are less convincing than other results in the paper. I find the regions of gene expression difficult to see in the figures. Without the arrows, I would not even have noticed the regions referred to. New experiments are needed to improve this, especially since the differences in gene expression are critical to the conclusions made. Can a color reaction that produces a color that would be more distinctive from background be done? If not an HCR or fluorescent in situ? I also am skeptical about effectively quantitating expression domains, used to generate panels C-F, when the signal is so weak and barely distinguishable from background. Finally, the number of animals is very small (4-6).

Minor comments

Line 105: “large” deletion mutations – actually these are relatively small deletions relative to the size of the genes.

Line 108: homeobox refers to the nucleic acid sequence and homeodomain refers to the protein sequence, so one or the other but not homeobox domain

Figure S1. I am not clear on how the bands on the gel relate to the sequences shown and how the gel provides evidence for the deletions. Please explain the genotyping process and Panel B in the Figure legend.

Generally a capitol H is used for Hox genes

**Have all data underlying the figures and results presented in the manuscript been provided?**

Reviewer #1: Yes

Reviewer #2: Yes

Reviewer #3: None

PLOS authors have the option to publish the peer review history of their article (what does this mean? ). If published, this will include your full peer review and any attached files.

**Do you want your identity to be public for this peer review?** For information about this choice, including consent withdrawal, please see our Privacy Policy .

Reviewer #1: **Yes:** Karen D. Crow

Reviewer #2: No

Reviewer #3: **Yes:** Leslie Pick

**Figure resubmission:**
---

## [Decision Letter · Decision Letter 1]

22 Jan 2026

PGENETICS-D-25-00736R1

Combined mutations of hoxa13a, hoxa13b, and hoxd13a lead to structural shifts in zebrafish soft fin rays providing insight into spiny ray evolution

PLOS Genetics

Dear Dr. Akimenko,

Thank you for submitting your manuscript to PLOS Genetics. After careful consideration, we feel that the revision was nicely responsive to prior review, but there are some points below that should be addressed prior publication. Therefore, we invite you to submit a revised version of the manuscript that addresses the points raised during the review process.

Please submit your revised manuscript within by Feb 21 2026 11:59PM. If you will need more time than this to complete your revisions, please reply to this message or contact the journal office at plosgenetics@plos.org. Please include the following items when submitting your revised manuscript:

We look forward to receiving your revised manuscript.

Kind regards,

Jeffrey J Essner

Academic Editor

PLOS Genetics

Pablo Wappner

Section Editor

PLOS Genetics

Aimée Dudley

Editor-in-Chief

PLOS Genetics

Anne Goriely

Editor-in-Chief

PLOS Genetics

**Journal Requirements:**

1) We have noticed that you have uploaded Supporting Information files, but you have not included a list of legends. Please add a full list of legends for your Supporting Information files after the references list.

**Reviewers' comments:**

Reviewer's Responses to Questions

**Comments to the Authors:**

Reviewer #2: Overall, the authors have improved the manuscript from their earlier submission and have addressed many, but not all, of the reviewers’ comments. I commend the authors for these improvement, including new experimental HCR in situ panels for quantifying Alx and Grem1b expression. These strengthen the results. However, several points still need to be addressed before publication.

The authors show that zebrafish mutants phenocopy spiny ray morphologies of acanthomorph fishes. The language in beginning in line 32 “our findings show that modifications to the expression of these hox genes likely played a large role in the evolutionary appearance of spines in fish fins” should be softened.

In the first part of the results (beginning line 126), the authors should collate the phenotype penetrance they describe in the text into frequency graphs to make it easier to follow.

In many instances, the text was difficult to follow. I would recommend additional editing feedback to improve readability prior to publication.

The structures highlighted by white and yellow arrow heads in Figure 1 still remain difficult to discern. As this is the first data shown, care should be taken to ensure the referenced anatomy is clear to the reader.

The features analyzed in Figure 4 should be specifically highlighted such that they clearly illustrates the summaries presented in Figure 3.

The distal margin of the zebrafish WT fin in Figure 6 A is cut off. The authors should try to use an image that includes the distal fin if the panels are meant to compare soft verse spiny ray morphologies.

It is generally recommended that all graphs should show individual data points for clarity.

**Have all data underlying the figures and results presented in the manuscript been provided?**

Reviewer #2: Yes

PLOS authors have the option to publish the peer review history of their article (what does this mean? ). If published, this will include your full peer review and any attached files.

**Do you want your identity to be public for this peer review?** For information about this choice, including consent withdrawal, please see our Privacy Policy .

Reviewer #2: No

**Figure resubmission:**
---

## [Editor Report · Decision Letter 2]

14 Feb 2026

Dear Dr Akimenko,

We are pleased to inform you that your manuscript entitled "Combined mutations of hoxa13a, hoxa13b, and hoxd13a lead to structural shifts in zebrafish soft fin rays providing insight into spiny ray evolution" has been editorially accepted for publication in PLOS Genetics. Congratulations!

Yours sincerely,

Jeffrey J Essner

Academic Editor

PLOS Genetics

Pablo Wappner

Section Editor

PLOS Genetics

Aimée Dudley

Editor-in-Chief

PLOS Genetics

Anne Goriely

Editor-in-Chief

PLOS Genetics

BlueSky: @plos.bsky.social

Comments from the reviewers (if applicable):

Thank you for hanging in there and improving the manuscript!

Jeff Essner

**Data Deposition**

http://datadryad.org/submit?journalID=pgenetics&manu=PGENETICS-D-25-00736R2

**Press Queries**

---

## [Editor Report · Acceptance letter]

PGENETICS-D-25-00736R2

Combined mutations of hoxa13a, hoxa13b, and hoxd13a lead to structural shifts in zebrafish soft fin rays providing insight into spiny ray evolution

Dear Dr Akimenko,

We are pleased to inform you that your manuscript entitled "Combined mutations of hoxa13a, hoxa13b, and hoxd13a lead to structural shifts in zebrafish soft fin rays providing insight into spiny ray evolution" has been formally accepted for publication in PLOS Genetics! Your manuscript is now with our production department and you will be notified of the publication date in due course.

With kind regards,

Anita Estes

PLOS Genetics

On behalf of:
